# Regionally distinct trophoblast regulate barrier function and invasion in the human placenta

**Bryan Marsh[1,2,3,4], Yan Zhou[1,3,5], Mirhan Kapidzic[1,3,5], Susan Fisher[1,3,5]\*, Robert Blelloch[1,2,3]\***

[1]The Eli and Edythe Broad Center of Regeneration Medicine and Stem Cell, University of California, San Francisco, San Francisco, United States; [2]Department of Urology, University of California, San Francisco, San Francisco, United States; [3]Center for Reproductive Sciences, University of California, San Francisco, San Francisco, United States; [4]Developmental and Stem Cell Biology Graduate Program, University of California, San Francisco, San Francisco, United States; [5]Department of Obstetrics, Gynecology, and Reproductive Sciences, University of California, San Francisco, San Francisco, United States

**\*For correspondence:**
susan.fisher@ucsf.edu (SF);
robert.blelloch@ucsf.edu (RB)

**Competing interest:** The authors declare that no competing interests exist.

**Abstract** The human placenta contains two specialized regions: the villous chorion where gases and nutrients are exchanged between maternal and fetal blood, and the smooth chorion (SC) which surrounds more than 70% of the developing fetus but whose cellular composition and function is poorly understood. Here, we use single cell RNA-sequencing to compare the cell types and molecular programs between these two regions in the second trimester human placenta. Each region consists of progenitor cytotrophoblasts (CTBs) and extravillous trophoblasts (EVTs) with similar gene expression programs. While CTBs in the villous chorion differentiate into syncytiotrophoblasts, they take an alternative trajectory in the SC producing a previously unknown CTB population which we term SC-specific CTBs (SC-CTBs). Marked by expression of region-specific cytokeratins, the SC-CTBs form a stratified epithelium above a basal layer of progenitor CTBs. They express epidermal and metabolic transcriptional programs consistent with a primary role in defense against physical stress and pathogens. Additionally, we show that SC-CTBs closely associate with EVTs and secrete factors that inhibit the migration of the EVTs. This restriction of EVT migration is in striking contrast to the villous region where EVTs migrate away from the chorion and invade deeply into the decidua. Together, these findings greatly expand our understanding of CTB differentiation in these distinct regions of the human placenta. This knowledge has broad implications for studies of the development, functions, and diseases of the human placenta.

## Editor's evaluation

By using single-cell RNA sequencing, elegant computational approaches, protein validation, and in vitro functional assays, this study characterizes the cellular composition and gene expression profiles of the human placenta in mid-gestation. In addition, this work gives new insights into our understanding of trophoblast differentiation in distinct regions of the human placenta. The findings and dataset provided by the authors represent an important resource for readers interested in human development and placenta biology.

## Introduction

The human placenta is the first organ to develop and forms the essential bridge between maternal and fetal tissues beginning at implantation (*Knöfler et al., 2019*; *Turco et al., 2018*). The placenta must begin development rapidly upon conception in order to perform the roles of future organ systems that have not yet developed and matured in the fetus, including nutrient and oxygen transport and protection from mechanical and pathogenic insults. The placenta also performs unique functions such as modulation of maternal tolerance and hormone production (*Maltepe and Fisher, 2015*; *Knöfler et al., 2019*; *Turco et al., 2018*). Placental development begins with the generation of stem villi surrounding the entire embryo and then proceeds asymmetrically to produce two distinct regions. At the human implantation site, the embryonic pole, the villi grow and branch to give rise to the region essential for the exchange of gases and nutrients, the chorion frondosum (also known as the chorionic villi or the villous chorion [VC]). This region includes the placental villi and the invasive EVTs. The villi located on the opposite side, the abembryonic pole, degenerate resulting in a smooth surface lacking villi termed the chorion leave (also known as the smooth chorion [SC]). The SC fuses with the amnion forming the chorioamniotic membranes (also known as the fetal membranes) (*Hamilton and Boyd, 1960*; *Boyd and Hamilton, 1967*; *Benirschke et al., 2006*).

Both the VC and SC are comprised of fetal derived cytotrophoblasts (CTBs), with CTBs in the VC differentiating to either multinucleate syncytiotrophoblast (STB) or to invasive extravillous trophoblast (EVT) (*Knöfler et al., 2019*). Compared to the VC, little effort has been made to analyze types and functions of the CTBs that comprise the SC (*Benirschke et al., 2006*; *Garrido-Gomez et al., 2017*). The CTBs of the SC exist in an epithelial-like structure and lack STBs and proximity to the fetal vasculature, and thus cannot function in a manner comparable to villi in VC (*Benirschke et al., 2006*). Furthermore, in contrast to the VC where EVTs invade and remodel the maternal arteries, the cells of the SC do not invade the adjacent decidua and the maternal blood vessels it contains (*Genbačev et al., 2015*). Thus, the function of these SC CTBs remains unclear.

Several pieces of evidence suggest that the SC is not simply a vestigial structure. First, the intact CTB layer contains proliferating cells and is maintained until term (*Yeh et al., 1989*; *Benirschke et al., 2006*; *Garrido-Gomez et al., 2017*). Second, the histological heterogeneity among SC CTBs suggests functional distinctions. *Yeh et al., 1989*, characterize two distinct populations of vacuolated and eosinophilic CTBs. Vacuolated CTBs were positive for placental lactogen and placental alkaline phosphatase, while the eosinophilic subpopulations was not. Both populations were rich in keratin and neither had the known characteristics of villous CTBs. *Bou-Resli et al., 1981* also note high levels of variation among CTBs in the SC and the existence of a vacuolated population. A more molecular characterization was carried out by *Garrido-Gomez et al., 2017*, which demonstrated heterogeneity of ITGA4 and HLA-G expression, markers previously associated with stemness and invasion, respectively (*Genbacev et al., 2016*; *McMaster et al., 1995*). This study also uncovered an expansion of the SC in cases of severe pre-eclampsia, along with a disease-specific gene expression pattern. In sum, these results suggest that the SC CTBs are a heterogeneous and dynamic collection of cells with important functions in development and disease.

Single cell RNA-sequencing (scRNA-seq) has emerged as the standard for transcriptional characterization of complex organs. This methodology was previously applied to the placenta, but with a focus on the maternal-fetal interface, specifically the chorionic villi and basal plate (*Liu et al., 2018*; *Suryawanshi et al., 2018*; *Vento-Tormo et al., 2018*). Recently, scRNA-seq was used to profile the smooch chorion at term (*Pique-Regi et al., 2019*; *Pique-Regi et al., 2020*; *Garcia-Flores et al., 2022*). However, in Pique-Regi et al., 2019, only 132 CTBs were identified in the SC out of 29,921 cells (0.44%) collected from this region. A comparable number of CTBs in the SC were recovered in *Garcia-Flores et al., 2022*, potentially reflecting a difficulty in capturing these cells at term. To better understand the differences in the cell types and functions of the two sides of the developing placenta, we applied scRNA-seq to matching samples of cells isolated from the VC and SC regions of human samples from mid to late in the second trimester. We used scRNA-seq to compare the composition and developmental trajectories of CTBs in the VC and SC. The data were validated and extended with functional studies to gain initial insights into the basis of differential migration of trophoblasts in each region. These results identified a novel SC-specific CTB population important for establishment of a protective barrier and the suppression of trophoblast invasion. In addition, these data represent a

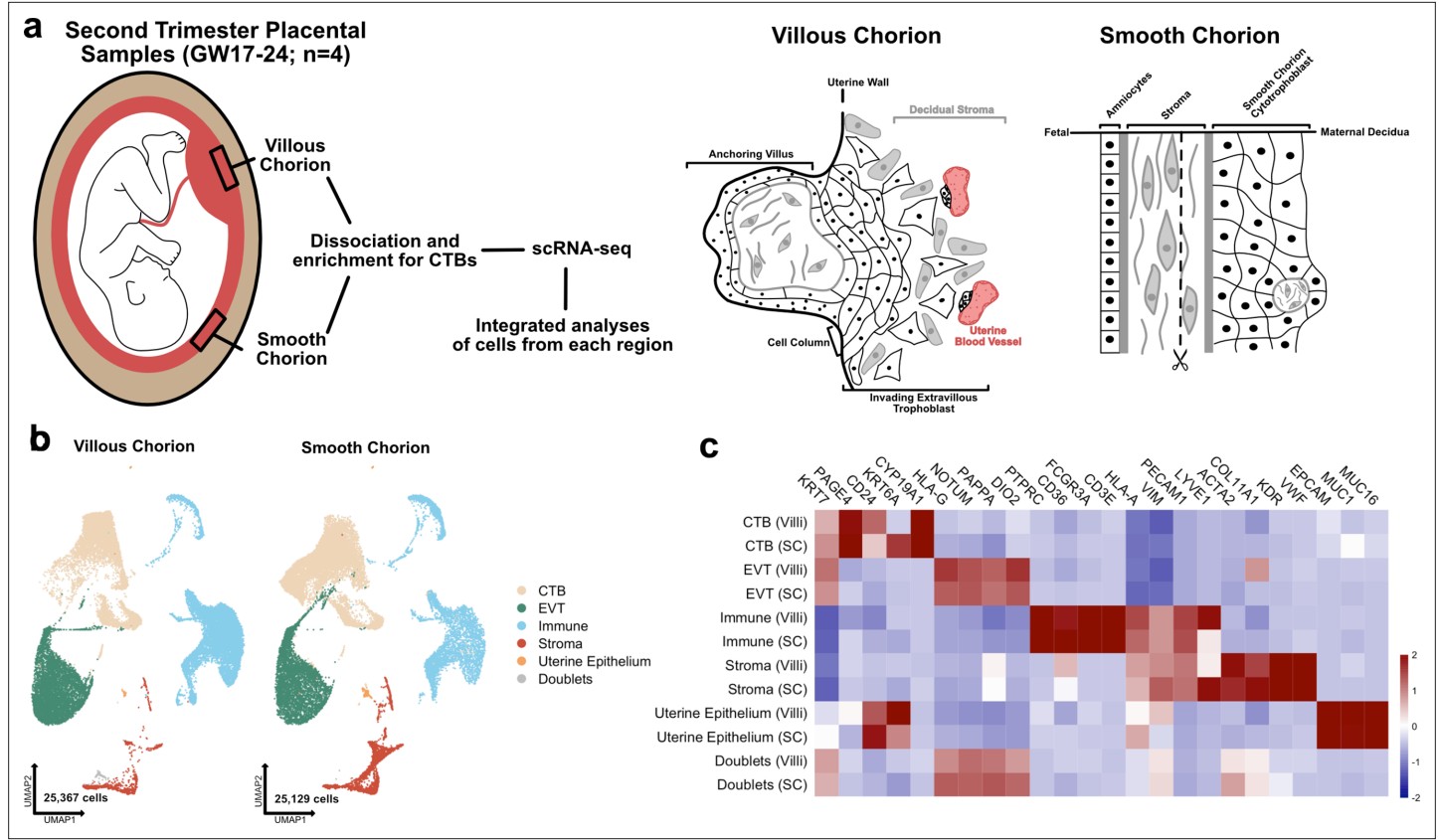

**Figure 1.** The transcriptional landscape of the villous (VC) and smooth chorion (SC) at mid-gestation. (**a**) Left: Schematic of the placenta at mid-gestation, highlighting the regions sampled, together with the methods used for cell isolation and characterization. Right: Schematic of the cell types and their organization in each region. (**b**) UMAPs of integrated samples, shown by region of origin (left – VC, right – SC), and colored according to broad cell type clusters. (**c**) Heatmap of the transcript expression of select cell identity markers across broad cell type clusters and regions. Values are scaled expression across the clusters of each region independently.

The online version of this article includes the following source data and figure supplement(s) for figure 1:

**Source data 1.** Marker genes for the clusters of the Immune subset.

**Source data 2.** Marker genes for the clusters of the Stroma subset.

**Figure supplement 1.** Metrics of the integrated dataset.

**Figure supplement 2.** *XIST* expression by sample.

**Figure supplement 3.** Comparison of stroma and immune clusters to *Vento-Tormo et al., 2018*.

**Figure supplement 4.** Comparison of stroma and immune clusters to *Pique-Regi et al., 2019*.

**Figure supplement 5.** Metrics and markers of the immune cell subset.

**Figure supplement 6.** Independent analysis of villous chorion (VC) and smooth chorion (SC) immune cells.

**Figure supplement 7.** Metrics and markers of the stromal cell subset.

**Figure supplement 8.** Independent analysis of villous (VC) and smooth chorion (SC) stromal cells.

resource of CTB types, proportions, and gene expression at mid-gestation against which age-related and pathogenic alterations can be measured.

## Results

### The transcriptional landscape of the VC and SC at mid-gestation

To understand the cellular composition of the SC, we isolated and profiled cells from both the VC and the SC regions of four second trimester human placentas spanning gestational weeks 18–24 (GW18–24) using scRNA-seq (*Figure 1a*). We chose to analyze second trimester samples because

the maturation of the SC is complete but the inflammation and apoptosis associated with membrane rupture and parturition is absent (*Benirschke et al., 2006*; *Yuan et al., 2006*; *Yuan et al., 2008*; *Yuan et al., 2009*; *Figure 1a*). SC and VC cells were isolated from each human placental sample allowing within and across patient comparisons. VC samples included cells isolated from floating and anchoring villi and areas surrounding the cell column, while most of the decidua (including spiral arteries) were dissected away. SC samples included the chorion and underlying stroma (mesenchymal and endothelial cells), but not the amnion and little of the neighboring decidua, which were also removed during dissection. CTBs were further enriched over stromal and immune cells during cell preparation as previously described (*Garrido-Gomez et al., 2017* and in Materials and methods). The transcriptomes of the resulting cells were captured using the 10× Genomics scRNA-seq platform.

Each of the eight datasets (GW17.6, 18.2, 23.0, 24.0; VC and SC) was captured independently, then integrated computationally (*Figure 1b*; *Stuart et al., 2019*). We classified the cells of the integrated dataset into broad cell type clusters according to functional identities and annotated each by expression of canonical markers. CTBs were annotated as *KRT7+*, *HLA-G-*; EVTs as *KRT7+*, *HLA-G+*; immune cells as *CD45+*, *VIM-*; stromal cells as *VIM+*, *CD45-*; and the uterine epithelium by expression of *EPCAM+*, *MUC1+*, and *MUC16+* (*Figure 1c*; *Lee et al., 2016*; *McMaster et al., 1995*; *Vento-Tormo et al., 2018*). Cells expressing exclusive markers of disparate cell types (co-expression of *KRT7, HLA-G, HLA-A, VIM, ACTA2*) were labelled as doublets and excluded from further analysis. The complete dataset used for further analysis contained 50,496 cells that passed quality control (between 500 and 6500 unique genes, fewer than 15% mitochondrial reads, doublets removed) (*Figure 1—figure supplement 1a*; *McGinnis et al., 2019*). Cells originating from each region (VC – 25,367 and SC – 25,129) and each sample (7181–17,705 cells per sample) were well represented (*Figure 1b*; *Figure 1—figure supplement 1b and c*). The number of cells in each broad cell type cluster demonstrated enrichment for CTBs, which represented more than 60% of the cells in the integrated dataset, as expected given the enrichment protocol (*Figure 1—figure supplement 1d*). The sex of each fetus was inferred by assaying expression of *XIST* in trophoblast cells isolated from each sample (*Figure 1—figure supplement 2a*), which allowed the assignment of cell types to either fetal or maternal origin (*Figure 1—figure supplement 2b*).

Even though there was enrichment for CTBs, we still identified 14,805 immune cells and 3883 stromal cells. We reclustered these immune and stromal cell subsets individually and compared them to previously published single cell analyses, allowing for the annotation of subtypes within each group (*Figure 1—figure supplement 3a-d*; *Figure 1—figure supplement 4a-d*; *Figure 1—figure supplement 5a-c*; *Figure 1—figure supplement 7a-c*; *Vento-Tormo et al., 2018*; *Pique-Regi et al., 2019*). Independent clustering analysis of VC and SC cells identified the similar populations as the integrated immune and stromal subsets and confirmed the relative proportions of cell identities across each region (*Figure 1—figure supplement 6a-d*; *Figure 1—figure supplement 8a-d*). All immune clusters robustly expressed *Xist* in each sample, identifying them to be of maternal origin (*Figure 1—figure supplement 2c*). Almost twofold more immune cells were recovered from the VC than the SC, although it is possible that this change in proportion is an artifact of dissection and/or the CTB enrichment protocol (*Figure 1—figure supplement 5d and e*). Comparing the immune cell types identified in each region revealed a higher proportion of macrophages in the VC as compared to the SC, which contained a greater proportion of NK/T cells (*Figure 1—figure supplement 5e*; *Figure 1—source data 1*).

While few stromal cells were isolated in the preparations, subclustering still revealed a differential composition of fetal stromal cells between the VC and SC (*Figure 1—figure supplement 7a-c*; *Figure 1—source data 2*). The majority of stromal cells recovered originated from the SC (2941 compared to 942 from VC). These cells included lymphatic endothelium (*Pique-Regi et al., 2019*) and two largely SC-specific mesenchymal cell populations of fetal origin, Mesenchyme 1 and Mesenchyme 3 (*Figure 1—figure supplement 7c-e*; *Figure 1—figure supplement 2b*). These two clusters are marked by elevated expression of *EGFL6*, *DLK1*, and uniquely by expression of *COL11A1*, which is observed only in the SC (*Figure 1—figure supplement 7b and f*). Interestingly, several canonical CTB support factors including HGF, WNT2, and RSPO3 were expressed in fetal stromal populations in both regions, suggesting shared requirements for WNT and MET signaling (*Figure 1—figure supplement 7g*). Taken together these data demonstrate the identification of broad classes of CTBs, immune, and support cells from both the VC and SC regions .

## Identification of an SC-specific CTB population

CTBs are the fetal cells that perform the specialized functions of the VC, and are required for normal fetal growth and development (*Maltepe and Fisher, 2015*; *Turco et al., 2018*; *Knöfler et al., 2019*). To better understand the composition of CTBs in the SC versus the VC, we subclustered this population (*KRT7+*, *VIM-*, *CD45-*, *MUC1-*). The CTB subset is comprised of 29,668 cells with similar representation and cell quality control metrics across all eight samples (*Figure 2—figure supplement 1a, b, and c*). This analysis identified 13 clusters including several CTB, EVT, and STB subtypes (*Figure 2a*; *Figure 2—source data 1*).

Broad classes of trophoblast were annotated by established markers (STBs – *CGA*, *CYP19A1*, *CSH1*, *CSH2*; EVTs – *HLA-G*, *DIO2*; CTBs – *PAGE4*, *PEG10*, and no expression of EVT and STB markers) (*Figure 2b* and *Figure 2—figure supplement 2a*; *McMaster et al., 1995*; *Lee et al., 2016*; *Suryawanshi et al., 2018*; *Liu et al., 2018*). Comparison to previously published cell types in the VC and SC confirmed the identities of most clusters (*Figure 2—figure supplement 1d-g*; *Vento-Tormo et al., 2018*; *Pique-Regi et al., 2019*). Two cell clusters showed high expression of canonical phasic transcripts, including *MKI67*, with the S-phase cluster denoted by expression of *PCNA* and the G2/M--phase cluster by expression of *TOP2A* (*Figure 2—figure supplement 2b*; *Tirosh et al., 2016*). Both populations share gene expression with all clusters of CTBs, and therefore, were identified as actively cycling CTBs (*Figure 2—figure supplement 2a*). No STB or EVT markers were identified in the cycling clusters as was expected due to the requirement for cell cycle exit upon terminal differentiation to these lineages (*Lu et al., 2017*; *Genbacev et al., 1997*).

CTBs separated into four clusters, CTB 1–4. CTB 1 cells highly expressed *PAGE4*, *PEG10*, and *CDH1* (*Figure 2b*, *Figure 2—figure supplement 2a*, *Figure 2—figure supplement 4a*), which have been shown to be canonical markers of villous CTB (*Lee et al., 2016*; *Suryawanshi et al., 2018*). Overall, CTB 2–4 were more transcriptionally similar to each other than to CTB 1, indicating a transcriptional program that was distinct from canonical villous CTBs (*Figure 2—figure supplement 2a* and c). CTB 2–4 existed along a gradient of gene expression changes suggestive of various stages of a common differentiation pathway. However, each population expressed distinct transcripts corresponding to important proposed functions of the SC. CTB 2 cells highly expressed *CLU*, *CFD*, and *IFIT3*, suggesting roles in responding to bacterial or viral infection through the innate arm of the immune system (*Thurman and Holers, 2006*; *Liu et al., 2011*). CTB 3 cells upregulated *EGLN3* and *SLC2A3*, indicating an HIF-mediated response to hypoxia and a switch toward glucose metabolism, likely as an adaptation to the decreased oxygen levels in the largely avascular SC region (*del Peso et al., 2003*; *Maxwell et al., 1997*). Finally, CTB 4 specifically expressed several cytokeratins — *KRT6A*, *KRT17*, and *KRT14*, found in many epithelial barrier tissues and important for maintenance of integrity in response to mechanical stressors (*Karantza, 2011*; *Figure 2b*, *Figure 2—figure supplement 2a*). Previous analysis of the SC region at term using scRNA-seq identified fewer than 0.5% of trophoblasts as being CTB and did not identify a separate KRT6 expressing CTB population (*Pique-Regi et al., 2019*). Computational integration of the data from *Pique-Regi et al., 2019*, with the trophoblast subset confirmed CTB 3 and CTB 4 to be unique to this study (*Figure 2—figure supplement 1f and g*). These results establish a previously underappreciated transcriptional diversity of CTB subpopulations.

Quantification of cells showed a strong regional bias in the number and proportion contributing to each CTB cluster (*Figure 2c and d*). In the VC samples, 53.8% of CTBs clustered in CTB 1 compared to 11.1% in the SC (*Supplementary file 1*). In contrast, the SC had a much larger proportion of the CTB 2–4 clusters. In this region, 71.3% of CTBs were nearly equally distributed among CTB 2 (25.6%), CTB 3 (24.9%), and CTB 4 (20.8%). In the VC, only 24.1% of CTBs were found in the same clusters, with the majority in CTB 2 (16.1%). The contribution to cycling clusters was consistent across regions: 22.1% and 17.5% of CTBs in the VC and SC, respectively. The relative proportions of CTB 1–4 in the VC and SC were consistent across individual samples indicating that this difference was not driven by sample variability (*Figure 2—figure supplement 1b-c*). Furthermore, independent clustering analysis of VC and SC cells identified the same populations as the integrated trophoblast subset. Importantly, independent clustering of each region did not identify CTB 4 in the VC or STB in the SC, suggesting CTB 4 and STB to be specific to the SC and VC, respectively. The small number of cells in the integrated dataset identified as VC CTB 4 (173 cells) or SC STB (14 cells) may be in part an artifact of computational integration (*Figure 2—figure supplement 3a-f*).

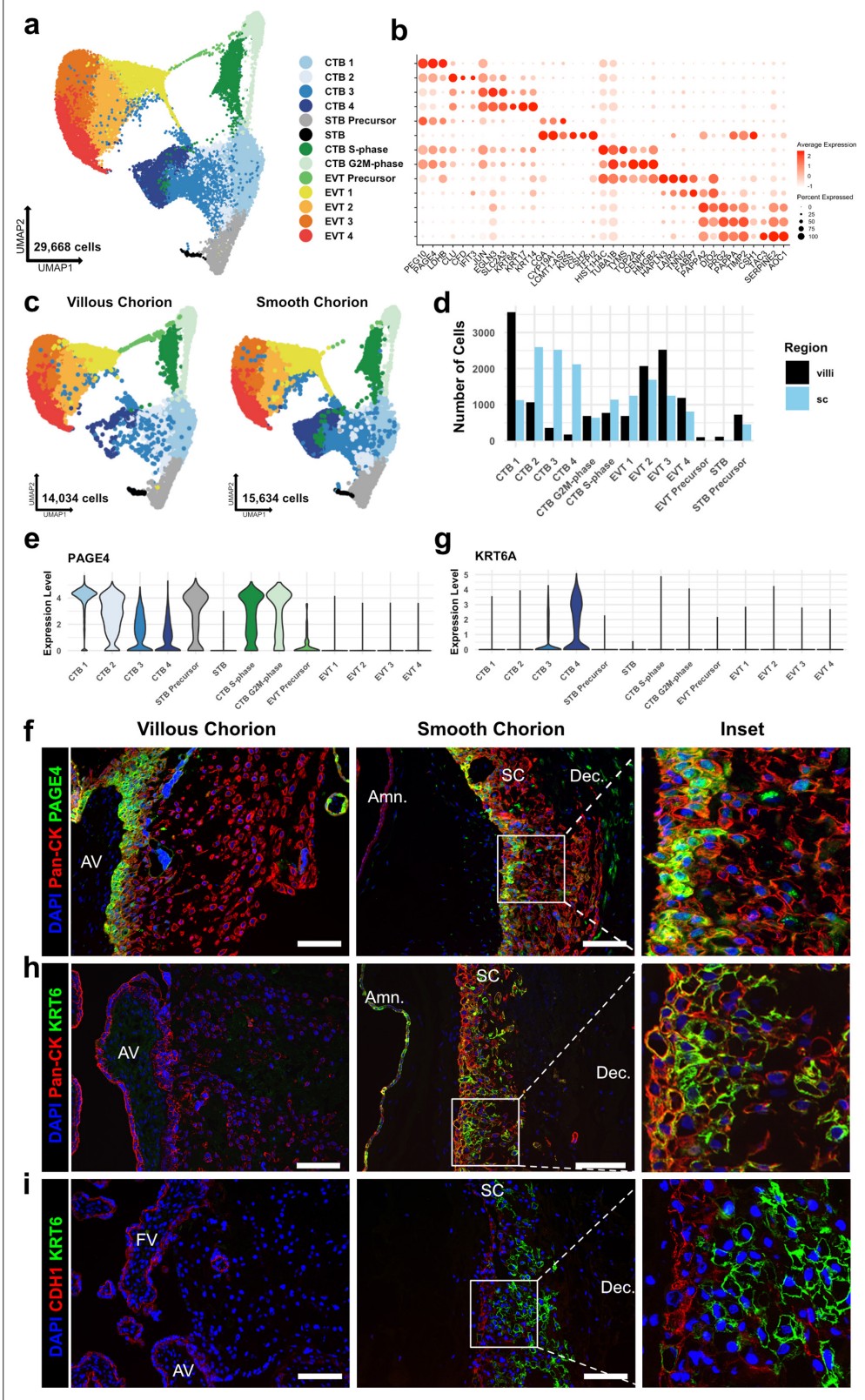

**Figure 2.** Identification of a smooth chorion-specific cytotrophoblast. (**a**) UMAP of subclustered trophoblasts (n=29,668). Colors correspond to the clusters at the right. (**b**) Dot plot showing average expression and percent of cells in each cluster as identified by the marker genes listed on the x-axis. The clusters are listed on the y-axis. (**c**) UMAP of subclustered trophoblasts from the villous chorion (VC) (left) or the SC (right). Clusters and colors are

*Figure 2 continued on next page*

*Figure 2 continued*

the same as in panel a. (**d**) Quantification of the number of cells in each trophoblast cluster from each region. Cells from the VC are shown in black. Cells from the SC are shown in blue. (**e**) Violin plot of *PAGE4* transcript expression across all trophoblast clusters. (**f**) Immunofluorescence co-localization of PAGE4 with pan-cytokeratin (marker of all trophoblast) in the VC (left) or SC (right). (**g**) Violin plot of *KRT6A* transcript expression across all trophoblast clusters. (**h**) Immunofluorescence co-localization of KRT6 with pan-cytokeratin (marker of all trophoblast) in the VC (left) or SC (right). (**i**) Immunofluorescence co-localization of CDH1 and KRT6 in the VC (left) or SC (middle). High magnification inset is denoted by the white box (right). For all images, nuclei were visualized by DAPI stain; scale bar = 100 μm. Abbreviations: AV = anchoring villi; FV = floating villi; SC = smooth chorion epithelium; Amn. = amnion; Dec. = decidua.

The online version of this article includes the following source data and figure supplement(s) for figure 2:

**Source data 1.** Marker genes for each cluster in the trophoblast subset.

**Figure supplement 1.** Metrics of the trophoblast subset.

**Figure supplement 2.** Markers of the trophoblast subset.

**Figure supplement 3.** Independent analysis of villous (VC) and smooth chorion (SC) trophoblast.

**Figure supplement 4.** KRT6 expression in the villous chorion (VC) region.

Next, we immunolocalized the protein products of genes that distinguished the subpopulations. At the mRNA level, *PAGE4* expression was highest in CTB 1 and decreased across CTB 2–4 (*Figure 2b and e*). In the VC, the CTB monolayer between the fetal stromal villous core and the overlying STB layer showed strong PAGE4 immunoreactivity, which diminished upon differentiation to EVT (*Figure 2f* - left). PAGE4 mRNA and protein expression matched that of known villous CTB marker CDH1 (*Figure 2—figure supplement 4a and b*; *Genbacev et al., 1997*). In the SC, the PAGE4 signal was strong in the epithelial layer directly adjacent to the fetal stroma and then decreased in cells distant from the basal layer, again matching CDH1 (*Figure 2f* – right, *Figure 2—figure supplement 4b* – right). Both RNA expression and protein localization were consistent with CTB 1 cells existing in both chorionic regions and occupying a similar niche.

Staining for KRT6, a marker highly enriched in CTB 4 cells (*Figure 2g*) showed a strikingly different result. Cells occupying the upper layers of SC epithelium showed a strong KRT6 signal, a pattern opposite to CDH1 (*Figure 2h and i* – right). KRT6 was absent from either the floating or anchoring villi of the VC (*Figure 2h and i* – left), although rare decidual resident KRT6 positive cells were identified in the VC region (*Figure 2—figure supplement 4c*, *Figure 6—figure supplement 1* – top). KRT6 isoforms, KRT6B and KRT6C, were not expressed, confirming KRT6A transcript and protein as highly specific markers of a CTB population found only in the SC (*Figure 2—figure supplement 4d*). These data describe a novel subpopulation of CTBs unique to the SC, which going forward we term CTB 4 or SC-CTBs for SC-specific CTBs.

## A common CTB progenitor gives rise to STBs in the VC and SC-CTBs in the SC

Next, we investigated the developmental origin of the SC-CTBs. We performed RNA velocity analysis to predict the relationships between cells based on the proportion of exonic and intronic reads. These predictions are shown as vectors representing both the magnitude (predicted rate) and the direction of differentiation (*Bergen et al., 2020*). We first asked whether RNA velocity could recapitulate the well-established differentiation trajectories of trophoblasts in the VC (*Knöfler et al., 2019*; *Turco et al., 2018*; *Vento-Tormo et al., 2018*). In accordance with previous results, RNA velocity projections identified CTB 1 as the root for three differentiation trajectories: self-renewal, differentiation to STBs, and differentiation to EVTs (*Figure 3a*). Cells at the boundary of the CTB 1 cluster showed differentiation vectors of high magnitude toward STB Precursors and upregulated canonical drivers of STB differentiation and fusion (*ERVW-1* and *ERVFRD-1*). These cells also expressed transcription factors (*GCM1* and *HOPX*) and hormones (*CSH1*) necessary for STB function (*Figure 3—figure supplement 1a*; *Baczyk et al., 2009*; *Mi, 2000*; *Blaise et al., 2003*; *Yabe et al., 2016*).

In the SC, CTB 1 cells once again were identified as the root for differentiation. However, CTB 1 cells showed strong directionality and magnitude toward CTB 2–4 (*Figure 3b*). All cells in CTB 1–4 clusters displayed uniform directionality indicating a robust differentiation trajectory ending at CTB

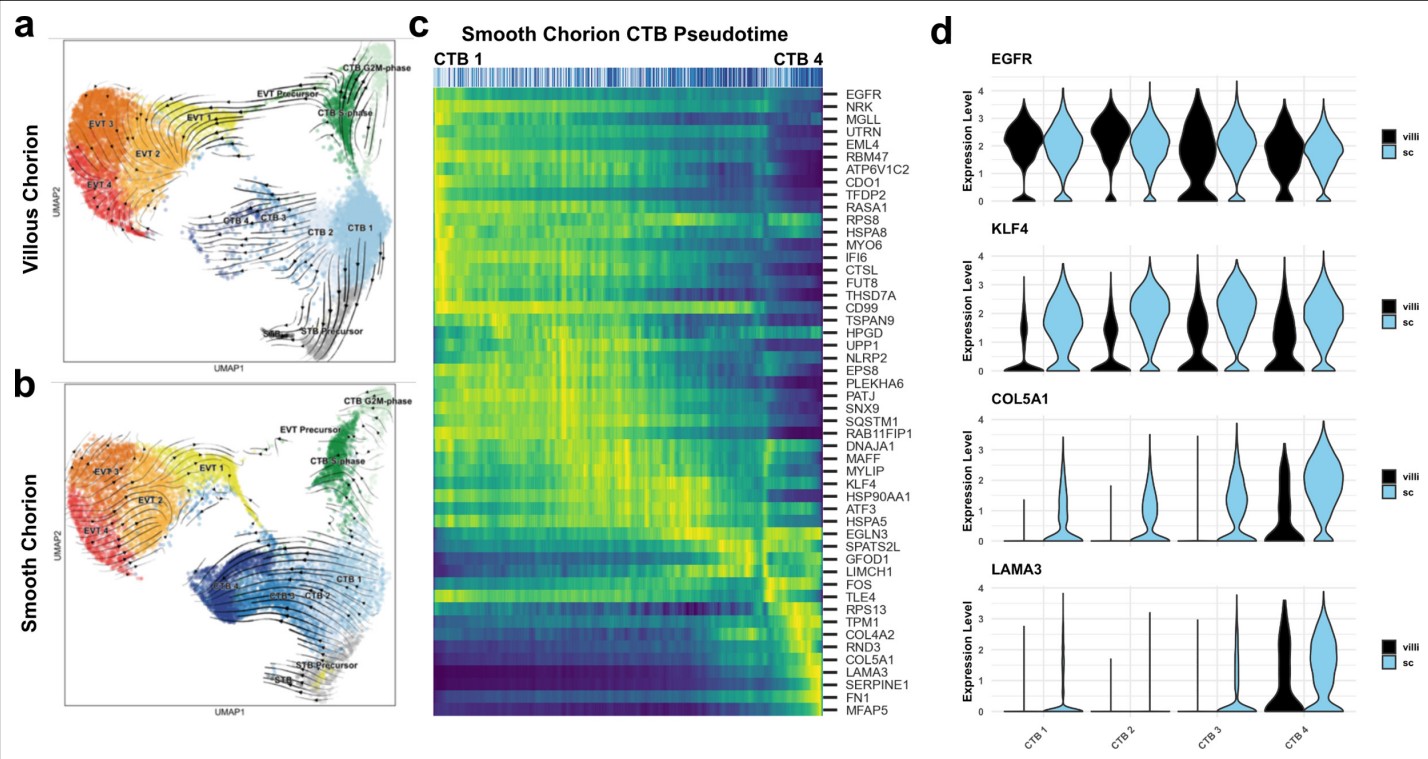

**Figure 3.** A common cytotrophoblast (CTB) progenitor gives rise to syncytiotrophoblasts (STBs) in the villous chorion (VC) and smooth chorion (SC)-CTBs in the SC. RNA velocity vector projections overlaid on to UMAPs for trophoblast cells isolated from the (**a**) VC and (**b**) SC. Arrows denote direction and magnitude is represented by line thickness. (**c**) Pseudotime reconstruction of SC derived CTB 1–4 clusters from the scVelo dynamical model of latent time. Each column represents one cell. Cells at the left are clustered in CTB 1 and progress through CTB 2, 3, and 4 along the x-axis. Select genes that were the major drivers of the pseudotime alignment are shown on the y-axis. Expression ranged from dark blue (lowest) to yellow (highest). (**d**) Violin plots of select factors from (**c**) demonstrated shared or region-specific expression for genes associated with the CTB 4 differentiation trajectory.

The online version of this article includes the following figure supplement(s) for figure 3:

**Figure supplement 1.** Smooth chorion (SC) trophoblast display reduced expression of syncytiotrophoblast (STB), increased expression of epithelial TFs, and proliferate.

**Figure supplement 2.** Predicted interactions between cytotrophoblasts (CTBs).

4. High levels of transcriptional similarity between CTB 2 and 4 compared to CTB 1 suggested CTB 2 and CTB 3 are intermediate states between CTB 1 and CTB 4 (*Figure 2—figure supplement 2c*). Mitotic KRT6+ cells were identified, and while the interaction between the cell cycle and differentiation of SC-CTBs remains unclear, these data show that differentiation to SC-CTB does not require cell cycle exit, unlike STBs and EVTs (*Figure 3—figure supplement 1b*). In contrast to the VC, we observed no velocity vectors with directionality toward the STB lineage from the CTB clusters in the SC samples. Further, the smaller number of STB precursors (460 cells) and STBs (14 cells) exhibited reduced expression of STB canonical markers such as ERVFRD-1 and GCM1, and notably, a near absence of ERVW-1 (*Figure 3—figure supplement 1a*). These cells may be associated with ghost villi (*Benirschke et al., 2006*). In sum, these data show differential developmental trajectories for the CTB 1 cells in the SC and VC, with the former largely giving rise to SC-CTBs and the latter to STBs.

To identify the genes that were correlated with progression from CTB 1–4 in the SC, we used the velocity vector predictions to construct a pseudotemporal model of differentiation. All the cells in these clusters were plotted in one dimension from the least to the most differentiated according the pseudotime model (*Figure 3c*). Genes that were highly expressed at the start of the pseudotemporal differentiation included pan-trophoblast factors such as *EGFR*, which was expressed throughout all four CTB populations in both the VC and SC. Progression along the pseudotime trajectory identified regulators of cell fate and function, including the transcription factor *KLF4* and extracellular matrix (ECM) components *COL5A1* and *LAMA3* (*Figure 3c–d*); all demonstrated SC-specific expression.

Elevated expression of ECM transcripts (*COL4A2*, *FN1*) and transcription factors responsive to cell contact and mechanical stress (*HES1*, *YAP1*) were coordinately upregulated, potentially highlighting the effects of the extracellular environment on fate specification (*Figure 3—figure supplement 1c*). To identify differential signaling events that might regulate alternative paths of differentiation in the VC and SC, we used CellPhoneDB to predict receptor-ligand interactions between CTB clusters within each region (*Figure 3—figure supplement 2a-b*; *Efremova et al., 2020*). This analysis identified BMP, Notch, and Ephrin signaling events specific to the SC region, which may help to determine cell fate and/or cell sorting within the SC trophoblast epithelium (*Figure 3—figure supplement 2b*). Together, these data demonstrated that SC-CTBs originate from CTB 1 progenitors common to both the VC and SC. In the SC, instead of upregulating syncytialization factors such as *GCM1* and *ERVFRD-1*, CTB 1 progenitors upregulate transcription factors such as *KLF4*, *YAP1*, and *HES1*, which drive an epithelial cell fate in other contexts (*Segre et al., 1999*; *Harvey et al., 2013*; *Rock et al., 2011*).

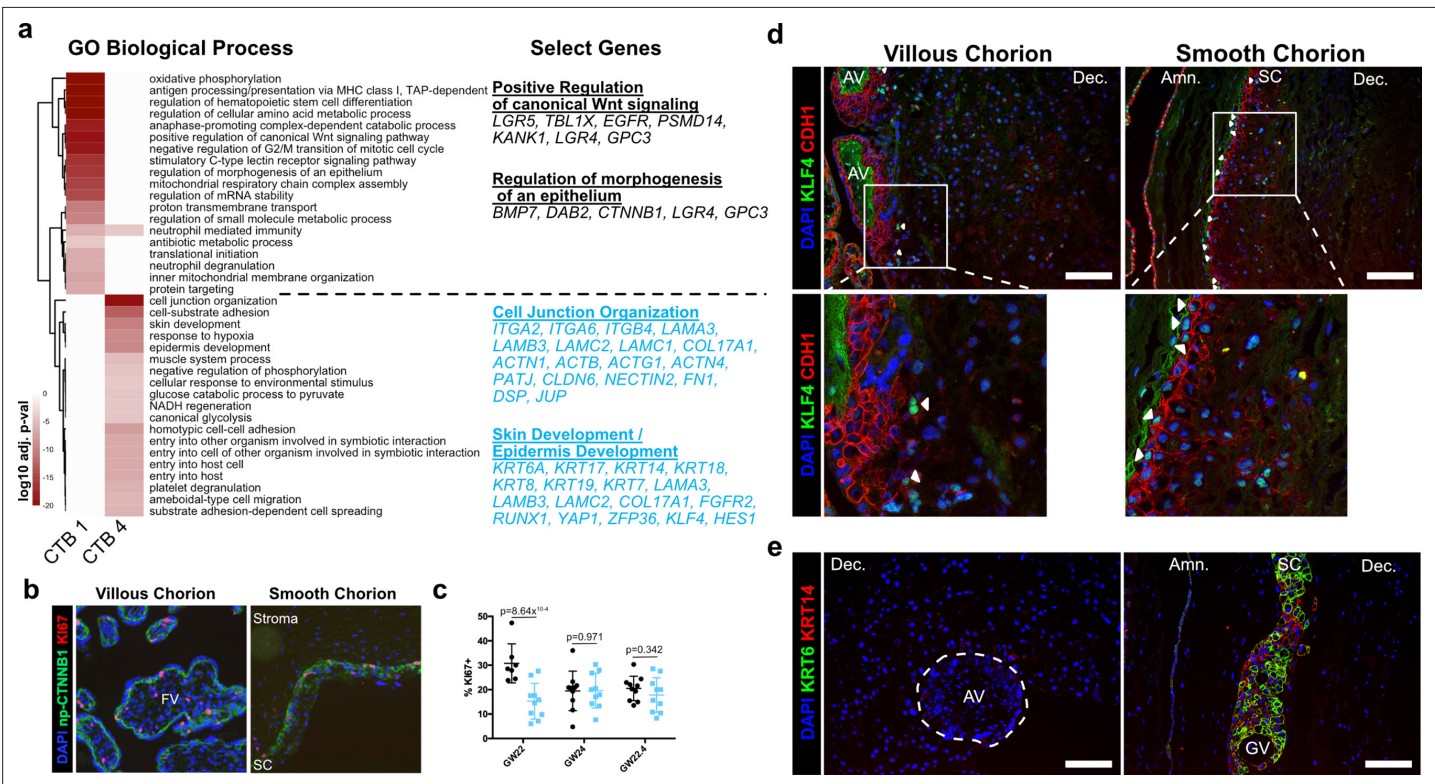

**Figure 4.** Smooth chorion-specific cytotrophoblasts (SC-CTBs) express a distinct epidermal transcriptional program. (**a**) Heatmap of gene ontology analysis adjusted p-values. Dark red corresponds to the lowest p-values and white represents p-values greater than 0.0005. Ontology categories are organized by hierarchical clustering along the y-axis. Marker genes for each cluster were used as inputs for the analysis. A subset of genes in selected categories are listed at the right. Categories and genes enriched in CTB 1 or CTB 4 are in black or blue, respectively. (**b**) Representative immunofluorescence co-localization of non-phosphorylated CTNNB1 and KI67 in the VC (left) and SC (right). (**c**) Quantification of the percent of np-CTNNB1 cells with KI67 expression in each region in three placental samples. Each dot represents the percentage in one field of view (at least seven per region per sample) as an estimate of mitotic cells per population. Percentages for the villous chorion (VC) region are shown in black and the SC region in blue. (**d**) Immunofluorescence co-localization of CDH1 and KLF4 protein in the VC (left) or SC (right). Arrowheads denote CDH1+/KLF4+ cells. (**e**) Immunofluorescence co-localization of KRT14 and KRT6 protein in the VC (left) or SC (right). The outline of the anchoring villi (AV) is denoted by the white dashed line. For all images, nuclei were visualized by DAPI stain; scale bar = 100 μm. Abbreviations: FV = floating villi; SC = smooth chorion epithelium; Amn. = amnion; Dec. = decidua; GV = ghost villi.

The online version of this article includes the following figure supplement(s) for figure 4:

**Figure supplement 1.** Complete cytotrophoblast (CTB) gene ontology analysis.

**Figure supplement 2.** Similarities and differences between cytotrophoblast (CTB) 1 in villous chorion (VC) and SC.

**Figure supplement 3.** Cytokeratin expression in villous (VC) and smooth chorion (SC) trophoblast.

**Figure supplement 4.** IFITM3 expression in cytotrophoblast (CTB) populations.

## SC-CTBs express a distinct epidermal transcriptional program

Next, we sought a better understanding of the physiological functions of the SC trophoblast clusters. We performed gene ontology analysis as a summary of functional processes (*Figure 4a*, *Figure 4— figure supplement 1*; *Yu et al., 2012*). We focused on the progenitor CTB 1 and terminally differentiated SC-CTBs as they showed enrichment for strikingly different functional categories. In CTB 1, we identified enrichment for WNT signaling, epithelial morphogenesis, and membrane transport, categories commonly associated with progenitors (*Figure 4a*, *Figure 4—figure supplement 1*). We validated the activity of WNT signaling and the location of these cells by immunolocalization of non-phosphorylated CTNNB1 (np-CTNNB1). Staining was localized to the most basal epithelial layer nearest to the stroma in both the VC and SC regions (*Figure 4b*), matching expression of the CTB 1 marker CDH1 (*Figure 4—figure supplement 2a*). WNT signaling has an important role in the maintenance of villous CTBs in vivo and in the derivation and culture of self-renewing human trophoblast stem cells (*Knöfler et al., 2019*; *Haider et al., 2018*; *Okae et al., 2018*). We investigated proliferation of np-CTNNB1 expressing cells in both regions using KI67 as a mitotic marker. This revealed a similar percentage of KI67+ CTB 1 cells, suggesting similar proliferative capacity across regions (*Figure 4b and c*). We next analyzed regional differences within CTB 1. Gene ontology identified an enrichment for oxidative phosphorylation and epithelial signaling cues in VC CTB 1 cells (*Figure 4—figure supplement 2b*). This is in direct contrast to SC CTB 1 cells that displayed elevated levels of hypoxia response genes (*Figure 4—figure supplement 2b-c*). KLF4 was identified in the RNA velocity analysis as gaining expression from CTB 1–4, but also showed greater expression in CTB 1 in the SC compared to the VC (*Figure 3d*). In accordance with the mRNA expression data, KLF4 protein often co-localized to CTB 1 in the SC, but was only observed in rare cells in the VC (*Figure 4d*). These data further support a similar location and function for CTB 1 in the SC and VC, with transcriptional and metabolic differences that presage distinct developmental trajectories.

CTB 4 ontological analysis strongly supported important roles for these cells in the formation of a protective barrier. The greatest enrichment was for cell junction and cell substrate adhesion genes that included numerous integrin and laminin subunits as well as junctional proteins *PATJ*, *DSP*, and *JUP*. An enrichment for both skin and epidermal development correlated with upregulation of the ECM and junctional transcripts. These categories included many cytokeratins (*KRT6A, 7, 8, 14, 17, 18*, and *19*) and transcription factors (*KLF4, YAP1*) required for epidermal identity and maintenance (*Segre et al., 1999*; *Schlegelmilch et al., 2011*). The organization of the SC is reminiscent of stratified epidermal cells of the skin, with progenitors adherent to the basal lamina and more differentiated cells progeny forming the upper layers. In many tissues, the specific domains of cytokeratin expression correspond to stratified cell layers with different functions. We asked if this was also the case in the SC epithelium. Cytokeratins 7, 8, and 18 were expressed in all trophoblast regardless of region (*Figure 4—figure supplement 3a*), but cytokeratin 6A, 14, and 17 displayed SC-specific expression that increased with differentiation toward CTB 4 (*Figure 4—figure supplement 3b*). Immunofluorescence localization confirmed expression of KRT14 as specific to the CTBs in the SC and inclusive of all KRT6 expressing cells (*Figure 4e*). These data support a model of accumulated cytokeratin expression that begins with CTB 1 (*KRT7, KRT8, KRT18*), increases in CTB 2–3 (*KRT14, KRT17*), and culminates in CTB 4 (*KRT6A*). Together these data are consistent with a central role for the CTBs of the SC in establishing a protective epithelial barrier for the rapidly growing fetus.

Beyond forming a physical barrier, important chorionic functions include protection from bacterial and viral infections. We identified an enrichment for genes involved in antiviral response in SC-CTBs, and in SC cells more broadly. For example, IFITM3, a restriction factor preventing entry of viruses into cells, is highly expressed in CTBs from the SC compared to the VC (*Figure 4—figure supplement 4*; *Bailey et al., 2014*; *Spence et al., 2019*). IFITM proteins have also been reported to inhibit syncytialization (*Buchrieser et al., 2019*), suggesting IFITM3 may also function to block differentiation of CTBs into STBs in the SC. Taken together, these data establish SC-CTBs as the building blocks and critical regulators of the SC barrier, responsible for both protection against physical forces and pathogen infection.

## EVTs of the VC and SC regions display distinct invasive activity but are transcriptionally similar

EVTs are the invasive trophoblasts of the placenta (*Knöfler et al., 2019*; *Turco et al., 2018*; *Red-Horse et al., 2004*). While the EVTs of the VC migrate away from the villi, invade the decidua, and replace the endothelial lining of the uterine arteries, the EVTs of the SC adhere to the decidua and do not home to the maternal vasculature (*Genbacev et al., 2016*). To understand the basis for these differences, we compared EVT subpopulations isolated from the VC and SC. Expression of the canonical marker of EVTs, *HLA-G*, was similarly abundant among the cells isolated from both chorionic regions (*Figure 5a and b*). A greater number (VC – 6572; SC – 5021) and larger proportion (VC – 46.83%; SC – 32.12%) of CTBs from the VC expressed HLA-G as compared to the analogous population from the SC (*Figure 5c*). Consistent with the transcript expression data, immunolocalization of HLA-G showed strong staining of cells in both the VC and SC. However, VC derived HLA-G positive cells were found deep in the maternal decidua. In contrast, HLA-G positive cells in the SC remained in the epithelial layer (*Figure 5b*).

EVTs were further divided into clusters 1–4. No clusters were specific to the VC or SC, except for the small number of putative EVT Precursors (VC – 99 cells; SC – 8 cells) (*Figure 5d*, *Figure 3a*). Previous work has classified EVTs in the VC as columnar, interstitial, or endovascular based on gene expression and location in the decidua (*Tilburgs et al., 2015*; *Knöfler et al., 2019*). The columnar subpopulation is believed to represent newly differentiated EVTs, which lie at the base of the columns that connect the anchoring villi to the uterine wall. Interstitial EVTs migrate through the decidua homing to maternal arteries, which they invade. The relationship between this subpopulation and endovascular EVTs that replace the maternal arterial endothelium is unclear, with evidence supporting endovascular EVTs arising either from interstitial EVTs and/or an independent origin (*Harris et al., 2009*; *Red-Horse et al., 2004*; *Pijnenborg et al., 2011*). We expected to capture both columnar and interstitial EVT populations, but few endovascular EVTs as the cellular preparations were largely devoid of arteries. We investigated the four clusters of EVT identified in our analysis in the context of the VC and the SC. Both regions contained cells from all clusters, although distinct regional biases were evident (*Supplementary file 1*). The SC region contained almost twice as many EVT 1 cells as the VC (VC – 684; SC – 1256), whereas the VC contained almost twice as many EVT 2–4 as the SC (VC – 5789; SC – 3757). We next asked which of the EVT clusters corresponded to known EVT classifications, indicating maturation state or invasive capacity. Gene ontology analysis showed an enrichment for placental development and antigen presentation categories in EVT 1 (*Figure 5—figure supplement 1a*). Representative marker genes for these categories in EVT 1 included several lineage-specific transcription factors including *GCM1*, *PPARG*, and *CEBPB* (*Figure 5—figure supplement 1b*; *Knöfler et al., 2019*; *Ferreira et al., 2016*). In contrast, EVT 2–4 showed enrichment for extracellular structure and matrix organization, glycosylation, and peptidase activity. EVT clusters 3–4 specifically showed increased expression for transcripts of proteases such as *HTRA1*, *MMP2*, and *MMP11* (*Figure 5—figure supplement 1c*). Based on the GO and specific gene enrichments, EVT 1 appeared most similar to columnar EVTs, while EVT 2–4 were consistent with interstitial EVTs that gain invasive capacity. The relative enrichment for EVT 1 in the SC (VC – 10.57% of EVT; SC – 25.05% of EVT) suggested an expansion in columnar-like EVTs at the expense of interstitial EVT. Conversely, the relative enrichment for EVT 2–4 in the VC region suggested an expansion of the invasive subpopulation (*Figure 5d*).

As almost 75% of EVTs from the SC region were in EVT clusters 2–4, we next asked if there were differences in gene expression between interstitial EVTs isolated from each region. This analysis identified relatively few differentially expressed genes, with the vast majority identified having a log fold change less than 1 (*Figure 5—figure supplement 2a*; *Figure 5—source data 1–4*). The small number of differentially expressed genes with a fold change greater than 1 was typically identified across multiple EVT clusters, suggesting an overriding transcriptional program that might supersede subcluster designations. To identify genes associated with this program, we combined all EVT subclusters into one cluster for analysis (*Figure 5e*). Differential expression analysis revealed the hormone CSH1 and several proteases including HTRA1, HTRA4, and PAPPA to be expressed at significantly higher levels in the more invasive VC EVTs. However, of the most differentially expressed transcripts, only CSH1, ITM2C, and DNASE1L3 were exclusive to the VC (*Figure 5—figure supplement 2b*). Immunolocalization of CSH1 confirmed this result (*Figure 5f*). CSH1, also known as human prolactin, is a secreted hormone that signals through the receptor

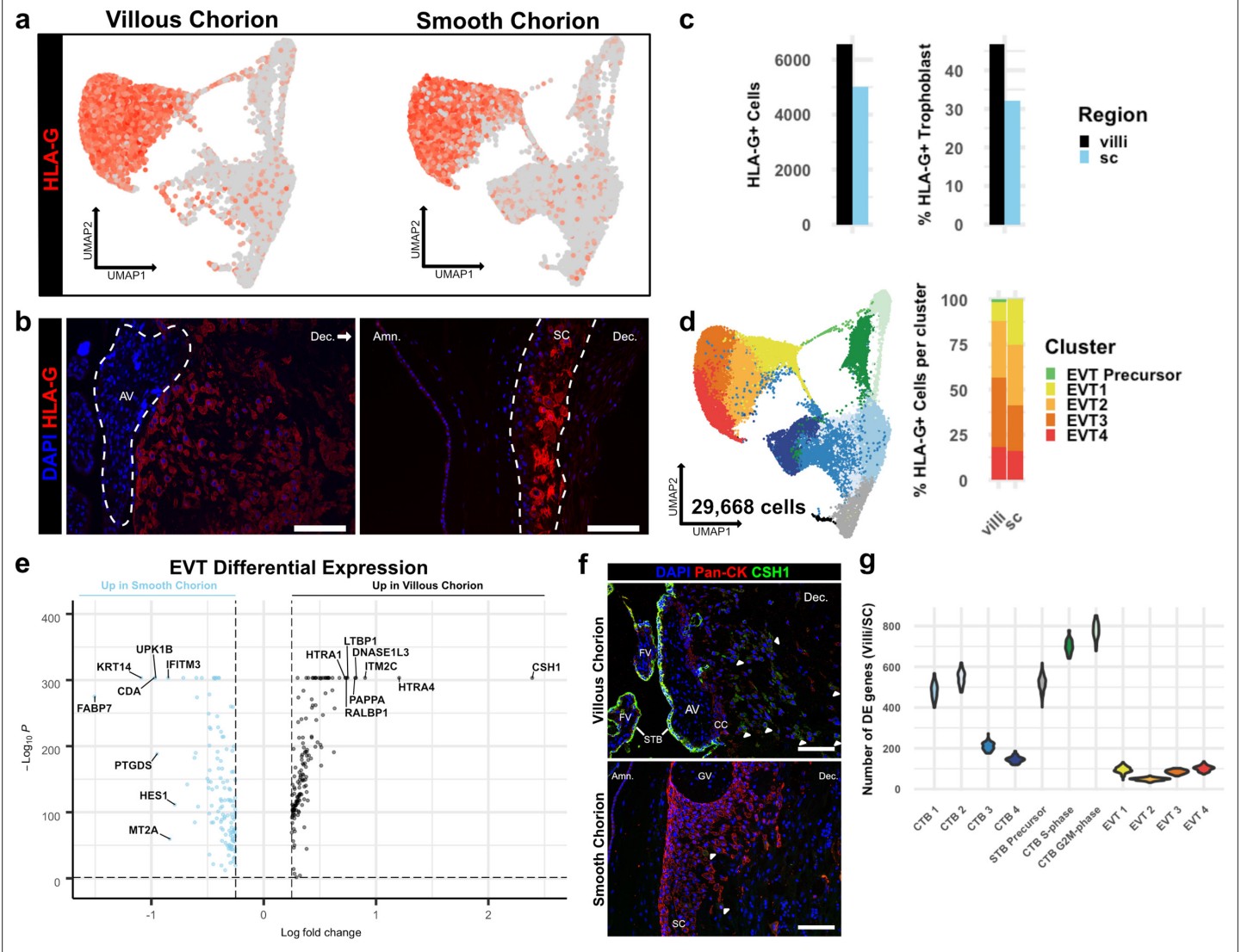

**Figure 5.** Extravillous trophoblasts (EVTs) of the villous chorion (VC) and SC regions display distinct invasive activity but are transcriptionally similar. (**a**) Expression of *HLA-G* transcript per cell projected in UMAP space for the VC (left) and SC (right). Expression ranged from low in light gray to high in dark red. (**b**) Immunofluorescence localization of HLA-G in the VC (left) and SC (right). The anchoring villi (AV) are outlined in white (left). The boundaries of the smooth chorion epithelium are denoted by the white lines (right). (**c**) Quantification of the number of *HLA-G* expressing extravillous trophoblast (EVT) (left) and the percent of total trophoblast that express *HLA-G* (right) for each chorionic region. (**d**) UMAP of all trophoblast cells including the EVT clusters (left). The percent of EVT cells in each cluster from each region (right). (**e**) Volcano plot of the differentially expressed genes between regions for all EVTs. All genes with a log fold change greater than an absolute value of 0.25 and a p-value of less than 0.05 were plotted. Those with greater expression in VC EVT are shown in black. Those with greater expression in SC EVT are shown in blue. (**f**) Immunofluorescence localization of CSH1 in the VC (top) and SC (bottom). Arrowheads denote CSH1 expressing cells. (**g**) Violin plots of the number of differentially expressed genes between 100 cells from each chorionic region within each cluster (100 permutations). Clusters with less than 100 cells per region were omitted due to the small sample size. For all images, nuclei were visualized by DAPI stain; scale bar = 100 μm. Abbreviations: FV = floating villi; SC = smooth chorion epithelium; Amn. = amnion; Dec. = decidua; STB = syncytiotrophoblast; CC = cell column.

The online version of this article includes the following source data and figure supplement(s) for figure 5:

**Source data 1.** EVT 1 cluster differentially expressed genes between regions.

**Source data 2.** EVT 2 cluster differentially expressed genes between regions.

**Source data 3.** EVT 3 cluster differentially expressed genes between regions.

**Source data 4.** EVT 4 cluster differentially expressed genes between regions.

**Figure supplement 1.** Functional annotation of extravillous trophoblast (EVT) clusters.

**Figure supplement 2.** Differential expression between extravillous trophoblasts (EVT) from villous (VC) and smooth chorion (SC).

**Figure supplement 3.** Predicted interactions between extravillous trophoblast (EVT) and stromal cells.

PRLR found on maternal cells. As such, the primary effect of CSH1 is likely non-cell autonomous, raising the possibility that the largest differences in EVTs between regions may not be in inherent invasive capacity, but rather their ability to communicate with the neighboring cells. Therefore, we asked whether differential EVT-stroma signaling interactions exist between the VC and the SC (*Efremova et al., 2020*). While this analysis underrepresents the contribution of the stroma due to its depletion by the trophoblast enrichment protocol, we identified enrichment for collagen-integrin interactions between EVT and stromal cells in the VC, which were not present in the SC (*Figure 5—figure supplement 3a*). In contrast, SC-specific EVT-stromal interactions included PDGFA and FN1 signaling (*Figure 5—figure supplement 3b*). As we see few transcriptional changes in EVTs between regions, our data support a model by which differential signaling may create a stromal environment more amenable to EVT invasion.

We next asked whether the gene expression differences between the VC and SC were smaller for EVT clusters than for all other trophoblast clusters. Calculating the Spearman correlation coefficient across each cluster, EVT 3, EVT 4, and CTB 1 (which we have established as common to both regions) were most similar and the only clusters with a coefficient greater than 0.75 (*Figure 5—figure supplement 2c*). Since this analysis does not account for cluster heterogeneity or size, we quantified the number of differentially expressed genes between 100 randomly selected cells from each region within each cluster. Across 100 permutations, the number of differentially expressed genes between regions was the lowest for all four EVT clusters (*Figure 5g*). Together, these results showed that the EVTs of the VC and SC are surprisingly similar even though they have distinct levels of invasion in vivo.

## CTBs of the SC inhibit EVT invasion

Given the transcriptional similarity between EVT clusters originating from the VC and SC, we wondered if a non-cell autonomous program could explain their distinct migratory properties. Immunofluorescence co-localization of CTB 1, SC-CTBs, and EVTs showed striking differences in the relative positioning of the EVTs and CTBs in the two regions (*Figure 6a and b*). On the VC side, which lacks SC-CTBs cells, HLA-G+ EVTs were distant from the np-CTNNB1+ CTB 1 cell population as expected (*Genbacev et al., 2016*). By the second trimester they had migrated away from the cell column of the anchoring villi into the decidua (*Figure 6a*). In contrast, HLA-G+ EVTs in the SC were adjacent to and in physical contact with KRT6+ SC CTBs (*Figure 6b*). The interactions between SC-CTBs and EVTs were numerous and widespread throughout the SC epithelium. Contacts between CTB 1 and EVTs in the SC were not observed and the VC contain only rare and irregular KRT6+ cells (*Figure 6—figure supplement 1*). This close association suggested possible paracrine signaling events between SC-CTBs and EVTs, which might impact each cell type.

To test this theory, we attempted to recapitulate the differential invasive properties of VC and SC EVTs in a transwell migration assay. Briefly, these cells were enriched using the same protocol as described for the scRNA-seq experiments. The cells were plated on Matrigel-coated transwell filters and cultured for 39 hr. Trophoblast projections that reached the underside of the filter, a proxy for invasion, were visualized by immunostaining with a pan-cytokeratin antibody (*Figure 6c*, *Figure 6—figure supplement 2*). Consistent with the differences observed in vivo, there was greater invasion of VC as compared to SC trophoblast (*Figure 6c*). Next, we asked whether CTBs from the SC region secreted soluble factors that inhibited invasion. The invasion assays were repeated with VC cells cultured with conditioned medium from SC cells and vice versa. While VC conditioned medium had no impact on either VC or SC cells, SC conditioned medium significantly reduced invasion of VC cells (*Figure 6d*). Neither conditioned medium impacted the density of VC or SC cells (*Figure 6—figure supplement 3*). Therefore, a secreted factor from SC cells inhibited the invasion of VC EVTs. Given that SC-CTBs were the only cell type unique to the SC side, it is highly likely that they produced the secreted factors that inhibit the invasion of EVTs to which they are juxtaposed. To identify which factors might be responsible for the repression of cellular invasion, we subset the predicted signaling interactions between CTB-EVT isolated from the SC (*Figure 6—figure supplement 4*). We then subset this analysis for only those containing secreted factors, which identified numerous interactions between CTB 4 and EVTs including both canonical cell signaling pathways (TNFa, PGF, TGFB1, FGF1, and PDGFB) and modifiers of the ECM (FN1 and THBS1) (*Figure 6e*). These data support a paracrine signaling mechanism by which CTB 4 cells restrict EVT invasion in the SC.

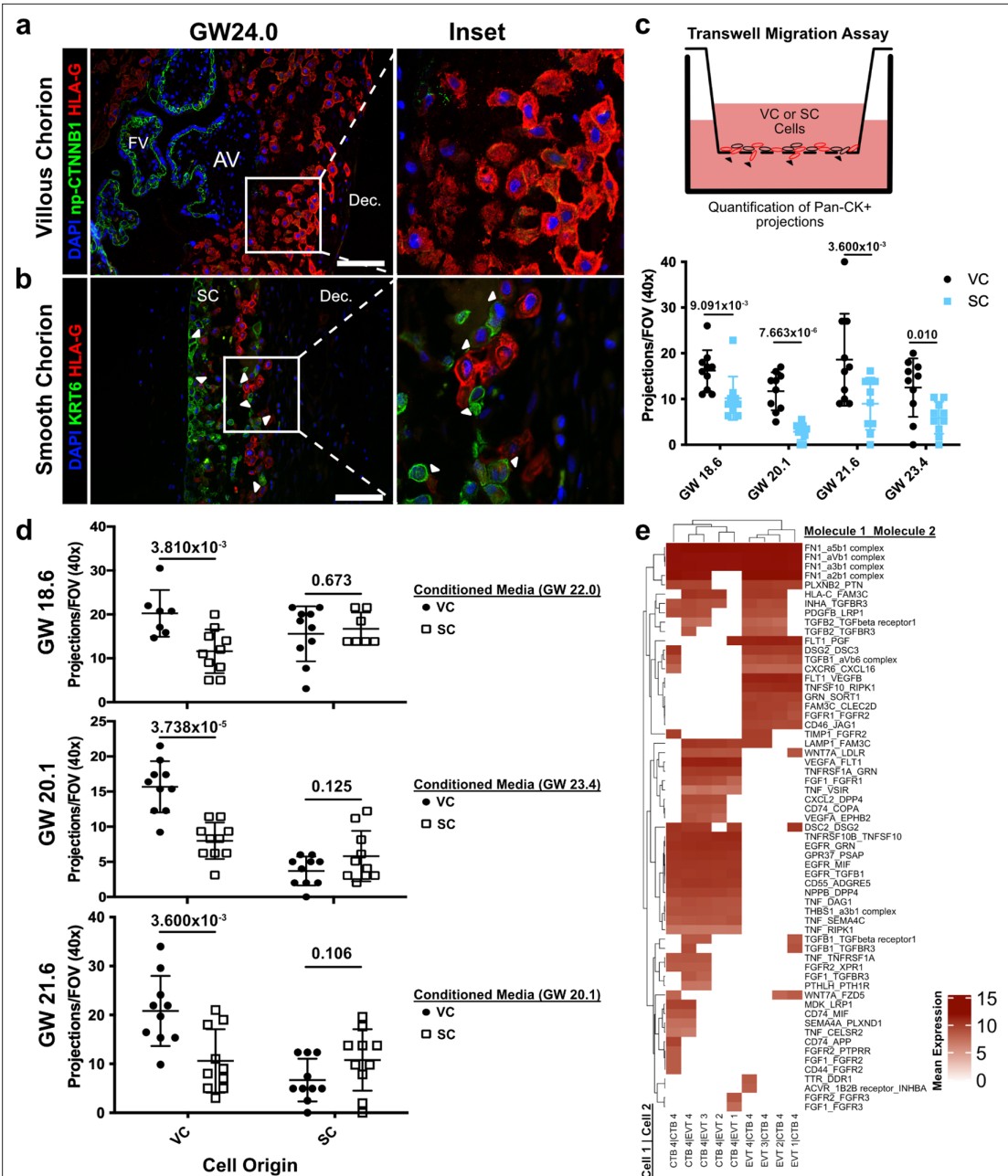

**Figure 6.** Cytotrophoblasts (CTBs) of the SC inhibit extravillous trophoblast (EVT) invasion. (**a**) Immunofluorescence co-localization of np-CTNNB1 and HLA-G in the VC. (**b**) Immunofluorescence co-localization of KRT6 and HLA-G in the SC. Arrowheads denote CTB and EVT interactions. (**c**) Schematic of the transwell invasion assay (top). Cells from either chorionic region were plated in the upper chamber of the transwell. After 39 hr of culture, the transwell membrane was fixed and stained with a Pan-cytokeratin antibody. The projections through the membrane are denoted by black arrowheads and quantified below. Results from the VC derived cells are shown in black, and the results from the SC derived cells are shown in blue. (**d**) Results from three biological replicates from each chorionic region cultured with conditioned medium from either VC or SC cells. The gestational ages of the plated cells are shown adjacent to the y-axis. The gestational ages of the cells from which conditioned medium was collected are noted in the legends at the right. The results for cells cultured with VC cell conditioned medium are denoted by black dots, and the results for those cultured in SC cell conditioned medium are denoted by open squares. p-Values were determined by t-test and are listed above each comparison. (**e**) Predicted receptor-ligand interactions from CellPhoneDB. The strength of interaction is estimated by mean expression and is plotted in the heatmap. Receptor-ligand interactions and cell pairs are listed such that Molecule 1 is expressed by Cell 1 and Molecule 2 is expressed by Cell 2. For all images, nuclei were visualized by DAPI staining; scale bar = 100 μm. Abbreviations: AV = anchoring villi; FV = floating villi; SC = smooth chorion epithelium; Dec. = decidua.

The online version of this article includes the following figure supplement(s) for figure 6:

**Figure supplement 1.** Cytotrophoblast (CTB)-extravillous trophoblasts (EVT) interactions in the villous chorion (VC) or SC region.

*Figure 6 continued on next page*

*Figure 6 continued*

**Figure supplement 2.** Representative images of the transwell invasion assay.

**Figure supplement 3.** Cell density is not correlated with culture in conditioned media.

**Figure supplement 4.** Complete predicted interactions between cytotrophoblast (CTB)-extravillous trophoblast (EVT) in the smooth chorion (SC).

## Discussion

This study profiled the developing second trimester placenta, providing a high-resolution molecular description of the trophoblast in the SC and placing them in context of the trophoblast from the VC of the same placenta. In doing so, we addressed long-standing observations concerning apparent similarities and differences between VC and SC trophoblast, deconstruct the trophoblast populations in the SC, and ascribe functions to the cells of this understudied region. Our study characterized a novel trophoblast population distinct from both extant CTB in the VC and from EVT found in either region. These trophoblasts reside only in stratified epithelium of the SC, express a specific cytokeratin (KRT6), and comprise at least 25% of all trophoblast in the region. Lineage reconstruction by RNA velocity suggested that the progenitors for these SC-specific CTBs are a resident proliferative progenitor trophoblast with a transcriptional profile that matches villous CTB. We identified several potential regulators of the transition from a common villous CTB-like cell to SC-CTB, including epidermal transcription factors such as KLF4 and YAP1. These molecules along with region-specific signaling events drive an epidermal transcriptional program creating the building blocks for a barrier against both physical and pathogen stressors. Finally, we identified CTB-EVT interactions occurring only in the SC and provide evidence that paracrine signaling between these populations restricts trophoblast invasion.

Since the SC is created by a degenerative process, consensus has been that these cells are remnants of this process, making the SC a vestigial structure (*Benirschke et al., 2006*). However, several previous observations suggested this might be an oversimplification. First, these cells were noted to be proliferative, suggesting either cellular expansion or turnover (*Benirschke et al., 2006*). We confirmed the presence of proliferative trophoblasts in the SC. Each cycling CTB cluster expressed markers of CTB 1–4 in both regions (*Figure 2—figure supplement 2a*) and both CTB 1 (*Figure 4b–c*) and SC-CTBs (*Figure 3—figure supplement 1b*) also expressed KI67 protein. Proliferation of both the progenitor and differentiated cells suggested a need for expansion coordinated with the growth of the developing fetus. Second, the importance of the SC was suggested by the noted heterogeneity of the trophoblasts within this region (*Yeh et al., 1989*; *Bou-Resli et al., 1981*; *Garrido-Gomez et al., 2017*; *Benirschke et al., 2006*). In agreement, our scRNA-seq results showed the SC to be a complex tissue with several trophoblast types. We identified, by differential transcription and function, at least two distinct CTB cell types, columnar and interstitial EVT, and a small number of STB Precursors. As discussed, all CTB and EVT populations either matched those found in the VC or had new region-specific functions. The only evidence of degeneration was the STB Precursor population within the SC, which lacked expression of both *ERVW-1* and *ERVFRD-1*, the fusogens necessary to form STB (*Mi, 2000*; *Blaise et al., 2003*; *Liu et al., 2018*). In sum, these studies suggested that the SC is a complex and functional tissue, not simply the remnant of villous degeneration.

We focused on CTB 4, a novel population found exclusively in the SC, in the context of the unique functions of this region. CTB 4 were enriched for epidermal, skin development, and antiviral gene categories suggesting concerted roles in the creation of a barrier against external forces and pathogens. Previous studies have noted that CTBs residing in the SC have high levels of keratin expression and are associated with extensive ECM deposits rich in laminin, collagen, and fibronectin (*Yeh et al., 1989*; *Bou-Resli et al., 1981*; *Benirschke et al., 2006*). Among all SC CTB populations, but most notably in CTB 4, we identified the upregulation of ECM components, including *LAMA3*, *COL5A1*, and *FN1*. Expression of ECM molecules in the SC is essential to protect against premature rupture of the fetal membranes, but its deposition has previously been ascribed to the stromal cells of the chorion (*Parry and Strauss, 1998*). Our results suggest that the trophoblasts of the SC create a specific ECM environment distinct from that of the VC. This altered composition likely has wide-ranging effects on trophoblast fate, gene expression, and activity. It will be interesting to explore the impact of advancing gestational age on the SC epithelium, especially with respect to the ECM. Presumably, cellular and structural changes precede the normal process of membrane rupture prior

to delivery at term. Whether premature rupture of the membranes phenocopies these events or is a unique process is an important open question.

Pathogen defense is another important function of the placenta. Infections of the chorion and amnion (chorioamnionitis) often result in adverse outcomes for mother and fetus, primarily preterm birth (**Romero et al., 2014**). We identified specific expression of the antiviral gene IFITM3 in CTB 4. IFITM3 expression is particularly interesting due to the recent finding that it blocks STB fusion mediated by the endogenous retroviral elements, ERVW-1 and ERVFRD-1. Therefore, IFITM3 expression in SC-CTBs may have a dual role – restricting viral entry into the cell and blocking the formation of STBs in the SC. Its expression also provides a mechanism by which the lack of STBs in the SC is maintained after degradation of the villi is complete.

A distinct feature of the CTBs of the SC was remodeling of the cytokeratin network coordinated with differentiation from CTB 1 to SC-CTBs (**Figure 2g**, **Figure 4—figure supplement 3b**). The progressive expression of KRT14, KRT17, and KRT6A is reminiscent of the cytokeratin code found in the epithelial cell layers of many tissues (**Karantza, 2011**). KRT14 is recognized as a marker of all stratified epithelium including subsets of basal progenitor and stem cells in several tissues, usually alongside KRT5 (**Moll et al., 1982**; **Nelson and Sun, 1983**; **Rock et al., 2009**). While we did not identify expression of KRT5 in any trophoblast, the KRT14 expressing proliferative cells of the SC fit the profile of stratified epithelia in other locations. The roles of KRT17 and KRT6 are less clear. Most knowledge about the function of these molecules comes from the epidermis in the context of injury and disease. KRT6 and KRT17 are upregulated rapidly upon epidermal injury and are expressed through the repair phase, each contributing specific functions (**Takahashi et al., 1998**; **McGowan and Coulombe, 1998**). KRT17 promotes proliferation and increases in cell size through Akt/mTOR and STAT3, respectively (**Kim et al., 2006**; **Yang et al., 2018**). KRT6 is a negative regulator of cell migration through the inhibition of Src kinase and associations with myosin IIA (**Wong and Coulombe, 2003**; **Rotty and Coulombe, 2012**; **Wang et al., 2018**). Finally, this molecule promotes expression of Desmoplakin and the maintenance of desmosomes, the latter, a long-established feature of SC trophoblasts (**Bou-Resli et al., 1981**; **Bartels and Wang, 1983**; **Benirschke et al., 2006**). Taken together, these observations are consistent with data on SC-CTBs. We found no evidence of their invasion, but did identify frequent interactions with the ECM, CTBs, and EVTs. The coordinated expression of these cytokeratins and the transcriptional profile writ large suggested that the SC has the properties of a highly specialized epidermis with a robust proliferative capacity and strong cohesive properties, but lacking migratory or invasive behavior.

As previous work has demonstrated or assumed, the majority of trophoblast identified are represented in both the VC and SC (**Benirschke et al., 2006**; **Garrido-Gomez et al., 2017**; **Pique-Regi et al., 2019**). We identified two cell types with striking transcriptional similarity between regions, CTB 1 and EVT. CTB 1 cells in the VC matched canonical villous CTB in transcriptional profile, activity, and niche. We identified a similar population in the SC. They are localized to an epithelial sheet juxtaposed to the fetal stroma, separated by a thin basal lamina. These cells are supported by similar signaling pathways in both regions, including HGF and WNT (**Figure 1—figure supplement 7g**; **Dokras et al., 2001**; **Okae et al., 2018**; **Zhou et al., 2002**). The similarities in location and growth factor requirements demonstrate that a niche similar to the villous trophoblast membrane exists in the SC. Additionally, we identified CTB 1 as the progenitor population for SC-CTB, and possibly for EVT in the SC. Thus, CTB 1 in both regions functions as multipotent progenitors. Despite the many similarities, the developmental trajectories emerging from CTB 1 differ in a region-specific manner. At present, it is unclear whether CTB 1 cells in both regions originally have the same potential or if fate restriction occurs as the fetal membranes form. Future experiments will need to functionally address whether CTB 1 from the VC can be coerced to generate SC-CTBs and if CTB 1 from the SC can efficiently generate STBs and invasive EVTs. Multipotent trophoblasts have been generated from first trimester SC cells, however, these cells have a distinct transcriptional profile from the second trimester cells we characterized (**Genbacev et al., 2016**).

Similarities between the EVT of each region, as defined by contact with the decidua, has been documented in observational, molecular, and most recently, transcriptomic studies (**Benirschke et al., 2006**; **Pique-Regi et al., 2019**). Our data confirmed a strong correlation between gene expression for all EVT populations, regardless of regional identity. Based on previously established EVT subtypes, we identified a reduction in the proportion of interstitial EVTs and a concomitant increase

in columnar-like EVTs of the SC. However, we did not find evidence of specific EVT subpopulations or intrinsic gene expression programs that would explain the differences in the depth of invasion of EVT in each region. However, we did uncover CTB-EVT interactions specific to the SC epithelium. Rather than physically separating as they do in the VC, CTBs and EVTs co-occupy the stratified epithelium of the SC, providing a contained environment for paracrine signaling, cell contact-mediated signaling, and cell-ECM interactions that may impact trophoblast invasion. In this study we show that soluble signaling factors from SC cells restrict the invasion of their counterparts from the VC. We can speculate on potential candidates that may function across multiple biological systems to restrict invasion. CTB 4 cells express high levels of TIMP1, TIMP3, and SERPINE1 (PAI-1), which inhibit the function and/or activation of MMPs necessary for trophoblast invasion (*Fisher et al., 1989*; *Zhu et al., 2012*). Addition of recombinant TIMPs or a function blocking antibody against plasminogen (whose processing is inhibited by SERPINE1) reduced trophoblast invasion in vitro (*Lala and Graham, 1990*). While these molecules are expressed by decidual cells, we identified their expression within the SC epithelium directly adjacent to EVTs. Another candidate regulator of invasion expressed by CTB 4 is TNFα. Treatment of trophoblasts with TNFα increases EVT apoptosis, decreases protease expression, increases SERPINE1 expression, and decreases invasion in vitro (*Otun et al., 2011*; *Xu et al., 2011*). These data suggest that SC-CTBs secrete multiple molecules that could decrease the survival and invasion of EVTs in the SC. In this study we explored the role of secreted factors, but SC-CTBs may also influence EVTs through cell contact-mediated signaling and the deposition of an ECM distinct from the basal plate. Future experiments will be necessary to explore the contributions of these distinct signaling pathways and matrices.

In summary, this study provides a high-resolution molecular accounting of the trophoblast that occupy SC. By comparison with trophoblasts isolated from the VC of the same placentas, we identified key similarities and differences to better understand the molecular determinants of trophoblast function and activity in the SC. We characterized a novel CTB population, marked by KRT6A expression, that is central to three key functions of the SC: formation of an epidermal-like barrier, blockage of aberrant STB differentiation, and restriction of EVT invasion. These data provide a better understanding of molecular and cellular pathways that control human placental development and against which disease-related changes can be identified and therapeutic targets discovered.

# Materials and methods

## Key resources table

| Reagent type (species) or resource | Designation | Source or reference | Identifiers | Additional information |
|---|---|---|---|---|
| Antibody | Rat anti E-cadherin Monoclonal Antibody (ECCD-2) | Thermofisher Scientific | 13-1900 | IF (1:250) |
| Antibody | Rabbit Anti-PAGE4 antibody (Polyclonal) | Sigma-Aldrich | HPA023880 | IF (1:100) |
| Antibody | Rabbit Recombinant Anti-Cytokeratin 6 Monoclonal antibody [EPR1603Y] | Abcam | ab52620 | IF (1:100) |
| Antibody | Rabbit anti Non-phospho (Active) β-Catenin (Ser33/37/Thr41) Monoclonal Antibody | Cell Signaling | 8814 | IF (1:100) |
| Antibody | Mouse anti Cytokeratin 14 Monoclonal Antibody (LL002) | Invitrogen | MA5-11599 | IF (1:100) with Antigen Retrieval |

*Continued on next page*

*Continued*

| Reagent type (species) or resource | Designation | Source or reference | Identifiers | Additional information |
|---|---|---|---|---|
| Antibody | Rabbit Anti-KLF4 antibody (polyclonal) | Sigma-Aldrich | HPA00292 | IF (1:100) |
| Antibody | Rat anti Cytokeratin (7D3) Monoclonal Antibody | Susan Fisher/University of California, San Francisco Cat# Fisher_001-clone7D3,RRID:AB_2631235 | AB_2631235 | IF (1:100) |
| Antibody | Mouse anti HLA-G (4H84) Monoclonal Antibody | Susan Fisher/University of California, San Francisco Cat# Fisher_002-clone4H84, RRID:AB_2631236 | AB_2631236 | IF (1:20) with Antigen Retrieval |
| Software, algorithm | R | https://www.r-project.org/ | | |
| Software, algorithm | ImageJ | ImageJ (http://imagej.nih.gov/ij/) | | |
| Software, algorithm | Seurat (3.1.3) | https://satijalab.org/seurat/ | | |
| Software, algorithm | cellranger (3.0.2) | https://support.10xgenomics.com/single-cell-gene-expression/software/pipelines/latest/feature-bc | | |
| Software, algorithm | ClusterProfiler | https://guangchuangyu.github.io/software/clusterProfiler/ | | |
| Software, algorithm | scVelo | https://github.com/theislab/scvelo (**Marsh, 2022b**) copy archived at swh:1:rev:1805ab4a72d3f34496f0ef246500a159f619d3a2 | | |
| Software, algorithm | Prism 6.0 | https://www.graphpad.com/scientific-software/prism/ | | |
| Software, algorithm | DoubletFinder | https://github.com/chris-mcginnis-ucsf/DoubletFinder (**Marsh, 2022c**) copy archived at swh:1:rev:67fb8b5808eb16167ead5f9b439677cc24837554 | | |

## Tissue collection

The University of California, San Francisco (UCSF) Institutional Review Board approved this study (11-05530). All donors gave informed consent. All samples are from elective terminations between gestational age of 17 weeks and 6 days and 24 weeks and 0 days.

## Cellular isolation of VC trophoblast

Trophoblast were isolated from both floating and anchoring villi dissected from second trimester human placentas. Trophoblast were isolated according to previously published protocols (*Fisher et al., 1989*; *Kliman and Feinberg, 1990*). Briefly, cuts were made at the base of each cotyledon near the chorionic plate and the entire villous tree up to and including the basal plate was taken for study. The decidua was removed from the basal plate side and the remaining villous tree was dissected and dissociated. The resulting floating and anchoring chorionic villi were washed in cold phosphate-buffered saline (PBS $Ca^{2+}$ and $Mg^{2+}$ free), dissected into 2–4 mm pieces, and filtered through a 1 mm mesh strainer to remove small pieces of tissue. CTBs were isolated from the tissue pieces by first removing the outer STB layer by collagenase digestion (Sigma-Aldrich; C-2674). Next, CTBs were dissociated by sequential enzymatic digestion (trypsin [Sigma-Aldrich; T8003; twice] and collagenase). Finally, CTBs were purified by Percoll density gradient centrifugation. Single cells were visually inspected for quality, counted using a hemacytometer, and immediately collected for scRNA-seq or for culture experiments.

## Cellular isolation of SC trophoblast

Trophoblast were isolated according to previously published protocols (*Garrido-Gomez et al., 2017*). Briefly, the fetal membranes were washed with PBS ($Ca^{2+}$ and $Mg^{2+}$ free) supplemented with 1% penicillin-streptomycin (10,000 units/ml penicillin; 10,000 µg/ml streptomycin), 0.003% fungizone (stock solution of 250 mg/ml), and 1% gentamicin. Next, the amnion and SC were manually separated and the amnion discarded. Next, the decidua parietalis was removed and discarded. The SC CTB layer

was then minced into small pieces (2–4 mm) and dissociated by sequential enzymatic digestion. First, the tissue pieces were incubated in PBS (10 ml/g of tissue) containing 3.5 mg collagenase, 1.2 mg DNase, 6.9 mg hyaluronidase, and 10 mg bovine serum albumin for 15–30 min. The supernatant was then discarded. Next, the tissue was incubated for 20–40 min in PBS containing trypsin (6.9 mg trypsin, 20 mg EDTA, 12 mg DNase per 100 ml; tissue weight: dissociation buffer volume = 1:8). The enzyme activity was quenched by adding an equal volume of media containing 10% FBS. The cell suspension was filtered through a 70 µm sterile strainer and centrifuged at 1200 $g$ for 7 min. A second collagenase digestion was performed by adding a 7× volume of the collagenase digestion buffer (see above), calculated on the basis of the weight of the cell pellet, followed by another incubation for 15–30 min. The cell suspension was then collected again by centrifugation. The cell pellets from the trypsin and second collagenase digestions were combined and purified over a Percoll density gradient centrifugation. Single cells were visually inspected for quality, counted using a hemacytometer, and immediately collected for scRNA-seq or for culture experiments.

## scRNA-seq and analysis

To capture the transcriptome of individual cells, we used the Chromium Single Cell 3' Reagent V3 Kit from 10× Genomics. For all samples 17,500 cells were loaded into one well of a Chip B kit for GEM generation. Library preparation including reverse transcription, barcoding, cDNA amplification, and purification was performed according to Chromium 10× V3 protocols. Each sample was sequenced on a NovaSeq 6000 S4 to a depth of approximately 20,000–30,000 reads per cell. The gene expression matrices for each dataset was generated using the CellRanger software (v3.0.2–10× Genomics). All reads were aligned to GRCh38 using STAR (https://support.10xgenomics.com/single-cell-gene-expression/software/pipelines/latest/advanced/references). The counts matrix was thresholded and analyzed in the package Seurat (v3.1.3). Cells with fewer than 500 or greater than 6000 unique genes, as well as all cells with greater than 15% mitochondrial counts, were excluded from all subsequent analyses. Doublet detection was performed for each sample using DoubletFinder and all doublets were excluded from analysis. For each sample, counts were scaled and normalized using ScaleData and NormalizeData, respectively, with default settings FindVariableFeatures used to identify the 2000 most variable genes as input for all future analyses. Principal component analysis (PCA) was performed using RunPCA and significant principal components (PCs) assessed using ElbowPlot and DimHeatmap. Dimensionality reduction and visualization using UMAP was performed by RunUMAP.

Integration of each timepoint into one dataset was performed using FindIntegrationAnchors and IntegrateData, both using 20 dimensions (after filtering each dataset for number of genes, mitochondrial counts, and normalizing as described above). Data scaling, PCA, selection of PCs, clustering and visualization proceeded as described above using 30 PCs and a resolution of 0.6.

To generate the trophoblast, stroma, and immune cell subsets, the respective clusters were subset from the integrated dataset using the function SubsetData based upon annotations from marker genes identified by FindAllMarkers. After subsetting, counts were scaled and normalized using ScaleData and NormalizeData, respectively, with default settings FindVariableFeatures used to identify the 2000 most variable genes. Differentially expressed genes for each integrated dataset were identified using FindAllMarkers.

## scVelo

RNA velocity analysis was applied to the entire conglomerate dataset using Velocyto to generate spliced and unspliced reads for all cells. This dataset was then subset for the trophoblast dataset introduced in *Figure 2*. The scVelo dynamical model was run with default settings and subset by each timepoint.

## Differential expression between regions

The number of trophoblast cells in each cluster were downsampled to 100 cells from each region of origin and performed differential expression using FindMarkers function and repeated this for 100 permutations. The number of genes found to be significantly differentially expressed (adj. p-value < 0.05) from each permutation are plotted. Clusters having less that 100 cells from each region were excluded from this analysis.

## Gene ontology analysis

Gene ontology analysis was performed with ClusterProfiler enrichGO function. The simplify function within this package was used to consolidate hierarchically related terms using a cutoff of 0.5. Terms were considered significantly enriched with an adjusted p-value of less than 0.05.

## Immunofluorescence staining

Placental tissue for cryosectioning was fixed in 3% PFA at 4°C for 8 hr, washed in 1× PBS, then submerged in 30% sucrose overnight at 4°C prior to embedding in OCT medium. Placental tissue in OCT was sectioned at 5 μm for all conditions. In brief, slides were washed in 1× PBST (1× PBS, 0.05% Tween-20), blocked for 1 hr (1× PBS +5% donkey or goat serum + 0.3% Triton-X), incubated in primary antibody diluted for 3 hr at room temperature (or overnight at 4°C), washed in 1× PBST, incubated in secondary antibody (Alexa Fluor 488 or 594) for 1 hr at room temperature, incubated in DAPI for 10 min at room temperature, washed in 1× PBST, and mounted and sealed for imaging. Any antigen retrieval was performed prior to the blocking step by heating the slides in a 1× citrate buffer with 0.05% Tween-20 at 95°C for 30 min. All antibodies and the dilutions are listed in the Key resources table. All immunofluorescence staining was performed in n=3 biological replicates (3 distinct placentas) and representative image is shown.

## Transwell invasion assay

Twenty-four-well plate transwell inserts with an 8 μM polycarbonate membrane (Corning Costar 3422) were coated with 10 μl of Matrigel (growth factor-containing, Corning Corp, Corning, NY) diluted 1:1 in Serum Free Media (95% DME H-21+ Glutamine, 2% Nutridoma (mostly β-D xylopyranose), 1% Pen Strep, 1% HEPES, 0.1% Gentamycin). Cells from each region of the placenta were isolated as described above and then plated in the upper well of the transwell insert at a density of 250,000 cells in 250 μl of Serum Free Media, with 1 ml of Serum Free Media (or conditioned media) in the well below the insert. The cells were then cultured for 39 hr. The cells in the transwell were then fixed in 3% PFA for 10 min at 4°C, permeabilized in ice-cold methanol for 10 min at 4°C, then washed in 1× PBS, incubated in Pan-CK primary antibody for 3 hr at 37°C, washed in 1× PBS, incubated with secondary antibody (Alexa Fluor 594) and DAPI for 1 hr at 37°C, washed in 1× PBS, then mounted and sealed for imaging. All experiments were performed in n=3 biological replicates (3 distinct placentas). For conditioned media experiments n=3 biological replicates were analyzed (cells and conditioned medium derived from 3 distinct placentas).

## Transwell invasion assay quantification

Transwell membranes mounted on slides were imaged at 40× magnification and the number of Pan-CK-positive projections through the membrane counted. Normalization for changes in cell density across fields of view and culture conditions was performed by quantifying the DAPI-positive area of each image and the number of projections normalized to the median across comparisons. Between 7 and 10 fields of view were quantified for each transwell membrane. p-Values were determined by t-test.

## Culture of placental cells and generation of conditioned media

Each well of a 24-well plate was coated with 20 μl of Matrigel undiluted (growth factor-containing, Corning Corp, Corning, NY) prior to cell seeding. Cells from each region of the placenta were isolated as described above and then plated into wells of a 24-well plate at a density of $1 \times 10^6$ cells in 1 ml of Serum Free Media (95% DME H-21+ Glutamine, 2% Nutridoma (mostly β-D xylopyranose), 1% Pen Strep, 1% HEPES, 0.1% Gentamycin). The cells were cultured for 39 hr. After 39 hr in culture the media was removed and centrifuged at 2000 $g$ for 10 min to remove cellular debris. The supernatant was then removed, snap frozen in LN2, and stored at –80°C. This media was then thawed and used as conditioned media.

## Data availability

All sequencing data is available at the NCBI Gene Expression Omnibus GSE198373. Processed data are available as R objects at https://figshare.com/projects/Regionally_distinct_trophoblast_regulate_barrier_function_and_invasion_in_the_human_placenta/135191. Code to process all raw data and

generate the datasets analyzed are available at https://github.com/marshbp/Regionally-distinct-trophoblast-regulate-barrier-function-and-invasion-in-the-human-placenta (*Marsh, 2022a*) copy archived at swh:1:rev:f8fb40282cc8d6c1bfabd4d4a06902eb92fefc94 . Previously published datasets from *Vento-Tormo et al., 2018 Pique-Regi et al., 2019*, are publicly available.

## Acknowledgements

We thank the members of the University of California – San Francisco National Center of Translational Research in Reproduction and Infertility for helpful comments during the design, execution, and publication of this project. We would also like to thank all members of the Blelloch and Fisher Labs for their comments and support. We specifically thank Ali San and Nasim Zeighami for their research contributions. We would also like to acknowledge our funding sources, the NIH Eunice Kennedy Shriver National Institute for Child Health and Human Development P50 HD055764 and NIH R37 HD076253.

## Additional information

### Competing interests

The authors declare that no competing interests exist.

### Funding

| Funder | Grant reference number | Author |
| --- | --- | --- |
| National Institutes of Health | P50 HD055764 | Bryan Marsh Robert Blelloch |
| National Institutes of Health | R37 HD076253 | Yan Zhou Mirhan Kapidzic Susan Fisher |

The funders had no role in study design, data collection and interpretation, or the decision to submit the work for publication.

### Author contributions

Bryan Marsh, Conceptualization, Software, Formal analysis, Validation, Investigation, Visualization, Writing – original draft, Writing – review and editing; Yan Zhou, Validation, Investigation; Mirhan Kapidzic, Investigation; Susan Fisher, Robert Blelloch, Conceptualization, Supervision, Funding acquisition, Writing – review and editing

### Author ORCIDs

Bryan Marsh http://orcid.org/0000-0002-4979-5233
Robert Blelloch http://orcid.org/0000-0002-1975-0798

### Ethics

Human subjects: The University of California, San Francisco (UCSF) Institutional Review Board approved this study (11-05530). All donors gave informed consent.

### Decision letter and Author response

Decision letter https://doi.org/10.7554/eLife.78829.sa1
Author response https://doi.org/10.7554/eLife.78829.sa2

## Additional files

### Supplementary files

• Supplementary file 1. Number of cells captured from each region per cluster for the trophoblast dataset.

• MDAR checklist

## Data availability

Sequencing data have been deposited in GEO under the accession code GSE198373 Processed data have been deposited on Figshare at https://figshare.com/projects/Regionally_distinct_tropho-blast_regulate_barrier_function_and_invasion_in_the_human_placenta/135191. Code to generate the processed data have been deposited on GitHub at https://github.com/marshbp/Regionally-distinct-trophoblast-regulate-barrier-function-and-invasion-in-the-human-placenta (copy archived at swh:1:rev:f8fb40282cc8d6c1bfabd4d4a06902eb92fefc94).

The following datasets were generated:

| Author(s) | Year | Dataset title | Dataset URL | Database and Identifier |
|---|---|---|---|---|
| Marsh BP, Blelloch RH, Fisher SJ | 2022 | Regionally distinct trophoblast regulate barrier function and invasion in the human placenta | https://www.ncbi.nlm.nih.gov/geo/query/acc.cgi?acc=GSE198373 | NCBI Gene Expression Omnibus, GSE198373 |
| Marsh BP | 2022 | Integrated_dataset.Robj | https://doi.org/10.6084/m9.figshare.19372868.v1 | figshare, 10.6084/m9.figshare.19372868.v1 |
| Marsh BP | 2022 | Stroma_subset.Robj | https://doi.org/10.6084/m9.figshare.19372532.v1 | figshare, 10.6084/m9.figshare.19372532.v1 |
| Marsh BP | 2022 | Immune_subset.Robj | https://doi.org/10.6084/m9.figshare.19372514.v1 | figshare, 10.6084/m9.figshare.19372514.v1 |
| Marsh BP | 2022 | Trophoblast_Subset.Robj | https://doi.org/10.6084/m9.figshare.19372466.v1 | figshare, 10.6084/m9.figshare.19372466.v1 |

The following previously published datasets were used:

| Author(s) | Year | Dataset title | Dataset URL | Database and Identifier |
|---|---|---|---|---|
| Vento-Tormo R | 2018 | Reconstructing the human first trimester fetal-maternal interface using single cell transcriptomics - 10x data | https://www.ebi.ac.uk/arrayexpress/experiments/E-MTAB-6701/ | ArrayExpress, E-MTAB-6701 |
| Pique-Regi R | 2019 | Single Cell Transcriptional Signatures of the Human Placenta in Term and Preterm Parturition | https://www.ncbi.nlm.nih.gov/projects/gap/cgi-bin/study.cgi?study_id=phs001886.v1.p1 | NIH dpGAP, phs001886.v1.p1 |

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
