## [Editor Report]

By using single-cell RNA sequencing, elegant computational approaches, protein validation, and in vitro functional assays, this study characterizes the cellular composition and gene expression profiles of the human placenta in mid-gestation. In addition, this work gives new insights into our understanding of trophoblast differentiation in distinct regions of the human placenta. The findings and dataset provided by the authors represent an important resource for readers interested in human development and placenta biology.

---

## [Decision Letter]

**Decision letter after peer review:**

Thank you for submitting your article "Regionally distinct trophoblast regulate barrier function and invasion in the human placenta" for consideration by *eLife*. Your article has been reviewed by 3 peer reviewers, one of whom is a member of our Board of Reviewing Editors, and the evaluation has been overseen by a Reviewing Editor and Didier Stainier as the Senior Editor. The reviewers have opted to remain anonymous.

Essential revisions:

Overall, this study is novel and addresses an important and uninvestigated question utilizing scRNA-seq and elegant computational approaches. The reviewers consider that the field will benefit from the publication of this research. Yet, there are caveats that the authors need to address before publication.

General comments:

(1) A point raised by both Reviewers #1 and #3 is about sample collection. We suggest the authors explain in detail in the method sections how the cellular isolations of VC and SC trophoblasts were performed and not just cite other papers. In addition, it would be good to know how the samples were obtained. Were they normal and voluntarily terminated? Reviewer #3 mentions that only very rarely do pregnancies fail in the second trimester which is a safe period for the mother and baby, so what is the reason for their delivery? If they were voluntary terminations of pregnancy they will have been exposed to mifepristone/misoprostol that will affect gene expression. Whichever way they were obtained they are not normal. Were the samples of the SC taken from close to the disc or near the cervix?

(2) Reviewers #1 and #2 raised a similar suggestion. The authors mention that there are differences in the EVTs isolated from the placental villi and those obtained from the chorioamniotic membranes. Other studies from the first and third trimesters have documented the presence of EVTs using scRNA-seq. I think that the study would benefit from comparing the authors' data (second trimester) with these publically available datasets. I think the authors could provide a unique roadmap of EVT transcriptomic activity throughout gestation.

(3) The findings on SC-CTB-secreted factors inhibiting EVT invasion are very intriguing. Could the authors analyze the conditioned media (via mass spec or ELISA) from SC and VC to try to identify potential candidates (e.g. SERPINE1 as mentioned in the discussion)?

(4) A major concern raised by Reviewer #2 is about the nomenclature used in the paper: villous chorion and smooth chorion. According to the schematic representation shown in Figure 1A, the authors sample the placental villous attached to the basal plate. Is this correct? It would have been more appropriate to sample the placental villous neighboring the chorionic plate or that from the inner placental mass, which is not associated with the basal or chorionic plate. In addition, the names utilized to define these placental compartments are unconventional. The readers would benefit from the authors utilizing conventional nomenclatures such as placental villi or chorioamniotic membranes. Last, the trophoblast in the chorioamniotic membranes is termed chorion laeve. The authors should consider using such terms throughout the manuscript and figures.

(5) Recent studies have suggested that stromal cells from the membranes participate in the host response against viral infection (COVID-19). Did the authors analyze the different types of stromal cell types between the placental villi and the membranes, and whether these display shared or unique functionality? Please discuss.

(6) Overall, the study would benefit from incorporating publically available data into the manuscript to strengthen the conclusions.

(7) Differences between CTBs and EVTs can also rely on cell-cell communications. Therefore, the authors are encouraged to perform Cell-Cell Chat analysis using their data.

(8) The manuscript is clearly presented and well written with a comprehensive reference to primary papers. However, many cited references are not included in the list including Maltepe 2015; Boll-Resli 1981; Pique-Regi 2020; Schlegelmilch 2011; Turco 2018; Kim 2006; Garcia-Flores et al. 2022.

(9) Reviewer #3 mentioned that identification of phenotypic markers specific to the smooth chorion is of interest and explains how this layer mediates important barrier functions for the fetus. An interesting finding is the specific expression of KRT6A in the smooth chorion. What is not clear from the report is how this layer is generated from the villi that originally surround the conceptus, which then regress probably due to the high oxygen concentrations at the periphery (Burton). This process is defective when the trophoblast transformation of vessels is abnormal in conditions, such as pre-eclampsia and preterm labour. The findings would be considerably more interesting if they could access samples earlier in pregnancy as the underlying pathogenesis of preterm labour is still unclear.

Thus, the important unanswered question is: what in these two microenvironments is driving the differential trajectory of CTB 1 towards villous SCT or to smooth KRT6A+ cells in the smooth chorion? Some TFs are different but what causes their upregulation in SC and this needs addressing? Are there any cells with the SC transcriptome in the new models of human TSC (Okae 2018; Turco 2018; Haider 2018)? These provide an in vitro model to generate SC and would considerably strengthen the paper.

(10) The methods of cell isolation rely on referring back to a previous publication and need expanding. "CTBs were enriched over stromal and immune cells – line 126" needs explanation as this will be a potential source of bias in the results. It appears that samples of the maternal side of the main villous placenta have been obtained – is this correct? Was decidua deliberately included? For the smooth chorion was the whole chorion – amnion, stroma, trophoblast and decidua included? These details are important as selecting specific areas and removing others will obviously alter the comparisons made.

(11) There are 6 macrophage populations – where are they located and which one is fetal Hofbauer cells? The relative paucity of NK cells and abundance of macrophages suggest that these are not from decidua. But why are there so few blood/decidual T cells? The best way to resolve these issues is to sequence maternal blood (and cord blood if obtainable) at the same time and remove these from the analyses.

To resolve these issues, the authors should compare their immune and fibroblast datasets in S2-S3 with previous single-cell data from the first trimester (Suryawanshi et al. 2018, Vento-Tormo et al. 2018). The correspondence between the decidual macrophages described in this manuscript is unclear compared with the ones from previous studies. Surprisingly, the 3 dNK cells previously identified are not found in this new dataset.

(12) The authors integrate the data from the villous chorion and smooth chorion computationally (eg Figure 1b). However, they should demonstrate that the integration pipeline is working correctly by defining the same cell states identified in the joint manifold when analysing the data separately. The same applies to all the analyses when specific cell subsets are zoomed in. For example, Figure 2c.

In addition, for each of the UMAPs defined in the manuscript, it is important to colour them by individual to confirm the integration.

(13) The EVT populations described are problematic and the logic of this section is hard to follow. Most trophoblast cells in the SC express the definitive EVT marker, HLA-G (Hutter 1996). So how do these EVT subsets relate to CTR2-4 and to published datasets? It is also surprising that the same EVT populations are defined in the analysis of villous and smooth chorion. An artifact due to the computational algorithm could be forcing the integration and it needs to be shown that this is not the case. This is common, as, by default, the method will assume equivalent populations in both datasets. One way to do so may be by using alternative methods for data integration (e.g., harmony, scVI), or by analysing datasets independently and comparing them afterwards (e.g. correlation analysis, machine-learning tools). It is also not made clear that the interstitial trophoblast deep in the decidua have not been sampled and only those EVT in the cell columns.

Detailed comments:

(1) Line 368 "within the arterial wall, this population (interstitial EVT) further differentiates to endovascular EVTs and replaces the maternal arterial endothelium (Harris 2009, Red-Horse 2004)". This is controversial with the majority view instead being that interstitial trophoblast mediates medial destruction and then there is the replacement of the endothelium by endovascular trophoblast moving down from the shell (Pijnenborg; Bulmer).

(2) Introduction: The second paragraph of the introduction could be rewritten and/or incorporated into the third paragraph. In its present form, it reads like a textbook. In addition, the introduction is somewhat too long and would benefit from some trimming.

(3) Introduction: The last paragraph of the introduction should be revised. The authors mention that previous studies using scRNA-seq were performed in the fetal membranes from term pregnancies (Page 3, Lines 104-105). Yet, no references were provided. In the following sentence (Page 3, Lines 105-106), the authors mention "in this study, CTBs were not identified in the smooth chorion…" and the authors cite four papers. Which of these papers are the authors referring to? In the 2019 study that is cited, the authors reported the presence of CTBs and EVTs in the fetal membranes. In addition, new studies have also reported the presence of CTBs and EVTs in the fetal membranes (PMIDs: 32662421, 35042863). The authors may consider including this information.

(4) Results, Page 3, Lines 122-124: The authors mention that they chose to analyze second-trimester samples to avoid inflammation and apoptosis associated with parturition. Couldn't the authors include samples from women who delivered at term without labor?

(5) Page 8, Lines 265-266: What is the evidence to suggest that the differentiation of SC-CTB does not require a cell cycle?

(6) Results, Page 5, Lines 157-159: The authors suggest the involvement of maternal immune cells in VC and SC; yet, these results are expected since both the placental villous and the chorioamniotic membranes (which were sampled in this study) represent the maternal-fetal interface: intervillous space and decidua parietalis, respectively.

(7) Results: The authors are congratulated for validating the mRNA results with protein expression. Specifically referring to Figure 2. The authors are also congratulated for testing functional differences between the EVTs from the placental villi and those from the membranes. I think that this is a very useful piece of information, and the functionality of the different subsets of EVTs requires further investigation.

(8) Results, Page 8, Line 245: The authors claim that the CTBs from the membranes and those from the placental villi share a common progenitor by performing an elegant RNA velocity analysis. Yet, wouldn't it be expected that all trophoblasts share a progenitor?

(9) Figure 5: What is the evidence that EVTs from the two different compartments are behaviorally distinct? In other words, what do the authors mean by "behavioral"?

(10) Immune cells Figure 1S2

How were the 4 different immune cell populations – maternal blood/decidual and fetal blood/placental cells distinguished? Both stromal and immune cells from both individuals will be present in these isolates. It is said by XIST that most of the immune cells are maternal – are they from blood or decidua? Were the fetuses all male? It would be better to look at the transcriptomes and deconvolute the genotype data by SNPs to confirm.

(11) Figure 1S3

The presence of lymphatics in decidua is disputed (Volchek 2010) and Hofbauer cells express LYVE1 (Bockle 2008) so how certain is the presence of lymphatics.

(12) In Figure 2 the authors introduce a CTB cycling population. Is this population corresponding to CTB-1 only? Or, is it a mix of CTB-1-4? If the latter, defining independent clusters by, for example, removing cell cycling genes from the highly variable genes considered to define the clusters should be performed.

(13) Line 417. The "suggesting a cell non-autonomous, rather than cell-autonomous, regulation of invasion" is speculation that should be included in discussion rather than the results. How do the authors explain the fact that trophoblast organoids invade in the absence of decidua as in tubal or abdominal pregnancy? To increase the impact the proteomes of the two conditioned media from VC and SC should be studied to identify what is inhibiting invasion into decidua.

*Reviewer #1 (Recommendations for the authors):*

– Line 368 "within the arterial wall, this population (interstitial EVT) further differentiates to endovascular EVTs and replaces the maternal arterial endothelium (Harris 2009, Red-Horse 2004)". This is a controversial statement. Please review well the literature on this and include additional references.

– It would be very interesting to know if there are SC-CTB present in the new in vitro models (trophoblast stem cells from Okae et al. 2018 or in the placental organoids obtained as in Turco et al. 2018 and Haider et al. 2018).

– It may be important to report the sex of the fetuses, especially in the sequencing data repository.

*Reviewer #2 (Recommendations for the authors):*

1. A major concern is the nomenclature utilized in this study: villous chorion and smooth chorion. According to the schematic representation shown in Figure 1A, the authors sample the placental villous attached to the basal plate. Is this correct? I think it would have been more appropriate to sample the placental villous neighboring the chorionic plate or that from the inner placental mass, which is not associated with the basal or chorionic plate. In addition, the names utilized to define these placental compartments are unconventional. The readers would benefit from the authors utilizing conventional nomenclatures such as placental villi or chorioamniotic membranes. Last, the trophoblast in the chorioamniotic membranes is termed chorion laeve. The authors should consider using such terms throughout the manuscript and figures.

2. Introduction: The second paragraph of the introduction could be rewritten and/or incorporated into the third paragraph. In its present form, it reads like a textbook. In addition, the introduction is somewhat too long and would benefit from some trimming.

3. Introduction: The last paragraph of the introduction should be revised. The authors mention that previous studies using scRNA-seq were performed in the fetal membranes from term pregnancies (Page 3, Lines 104-105). Yet, no references were provided. In the following sentence (Page 3, Lines 105-106), the authors mention "in this study, CTBs were not identified in the smooth chorion…" and the authors cite four papers. Which of these papers are the authors referring to? In the 2019 study that is cited, the authors reported the presence of CTBs and EVTs in the fetal membranes. In addition, new studies have also reported the presence of CTBs and EVTs in the fetal membranes (PMIDs: 32662421, 35042863). The authors may consider including this information.

4. Results, Page 3, Lines 122-124: The authors mention that they chose to analyze second-trimester samples to avoid inflammation and apoptosis associated with parturition. Couldn't the authors include samples from women who delivered at term without labor?

5. Results, Page 5, Lines 157-159: The authors suggest the involvement of maternal immune cells in VC and SC; yet, these results are expected since both the placental villous and the chorioamniotic membranes (which were sampled in this study) represent the maternal-fetal interface: intervillous space and decidua parietalis, respectively.

6. Results: The authors are congratulated for validating the mRNA results with protein expression. I am specifically referring to Figure 2.

7. Results, Page 8, Line 245: The authors claim that the CTBs from the membranes and those from the placental villi share a common progenitor by performing an elegant RNA velocity analysis. Yet, wouldn't it be expected that all trophoblasts share a progenitor?

8. Page 8, Lines 265-266: What is the evidence to suggest that the differentiation of SC-CTB does not require a cell cycle?

9. The authors mention that there are differences in the EVTs isolated from the placental villi and those obtained from the chorioamniotic membranes. Other studies from the first and third trimesters have documented the presence of EVTs using scRNA-seq. I think that the study would benefit from comparing the authors' data (second trimester) with these publically available datasets. I think the authors could provide a unique roadmap of EVT transcriptomic activity throughout gestation.

10. Figure 5: What is the evidence that EVTs from the two different compartments are behaviorally distinct? In other words, what do the authors mean by "behavioral"?

11. I would like to congratulate the authors for testing functional differences between the EVTs from the placental villi and those from the membranes. I think that this is a very useful piece of information, and the functionality of the different subsets of EVTs requires further investigation.

12. Recent studies have suggested that stromal cells from the membranes participate in the host response against viral infection (e.g., COVID-19, PMID: 35042863). Did the authors analyze the different types of stromal cell types between the placental villi and the membranes, and whether these display shared or unique functionality? Please discuss.

13. Overall, I think the study would benefit from incorporating publically available data into the manuscript to strengthen the conclusions.

14. Last, differences between CTBs and EVTs can also rely on cell-cell communications. Therefore, the authors are encouraged to perform CellChat analysis using their data.

*Reviewer #3 (Recommendations for the authors):*

This is an analysis of the smooth chorion compared to the invasive trophoblast populations in the second trimester of human pregnancy.

The manuscript is clearly presented and well written with a comprehensive reference to primary papers. However, many cited references are not included in the list including Maltepe 2015; Boll-Resli 1981; Pique-Regi 2019; Schlegelmilch 2011; Turco 2018; Kim 2006.

The identification of phenotypic markers specific to the smooth chorion is of interest and explains how this layer mediates important barrier functions for the fetus. An interesting finding is the specific expression of KRT6A in the smooth chorion. What is not clear from the report is how this layer is generated from the villi that originally surround the conceptus which then regress probably due to the high oxygen concentrations at the periphery (Burton). This process is defective when the trophoblast transformation of vessels is abnormal in conditions such as pre-eclampsia and preterm labour. The findings would be considerably more interesting if they could access samples earlier in pregnancy as the underlying pathogenesis of preterm labour is still unclear.

Thus, the important unanswered question is: what in these two microenvironments is driving the differential trajectory of CTB 1 towards villous SCT or to smooth KRT6A+ cells in the smooth chorion? Some TFs are different but what causes their upregulation in SC and this needs addressing? Are there any cells with the SC transcriptome in the new models of human TSC (Okae 2018; Turco 2018; Haider 2018)? These provide an in vitro model to generate SC and would considerably strengthen the paper.

How were the samples were obtained?

No details are given about the clinical details of these pregnancies. Were they normal and voluntarily terminated? Pregnancies only very rarely fail in the second trimester which is a safe period for the mother and baby so what is the reason for their delivery?

If they were voluntary terminations of pregnancy they will have been exposed to mifepristone/misoprostol that will affect gene expression. Whichever way they were obtained they are not normal.

Were the samples of the SC taken from close to the disc or near the cervix?

The methods of cell isolation rely on referring back to a previous publication and need expanding. "CTBs were enriched over stromal and immune cells – line 126" needs explanation as this will be a potential source of bias in the results. It appears that samples of the maternal side of the main villous placenta have been obtained – is this correct? Was decidua deliberately included? For the smooth chorion was the whole chorion – amnion, stroma, trophoblast and decidua included? These details are important as selecting specific areas and removing others will obviously alter the comparisons made.

Immune cells Figure 1S2

How were the 4 different immune cell populations – maternal blood/decidual and fetal blood/placental cells distinguished? Both stromal and immune cells from both individuals will be present in these isolates. It is said by XIST that most of the immune cells are maternal – are they from blood or decidua? Were the fetuses all male? It would be better to look at the transcriptomes and deconvolute the genotype data by SNPs to confirm.

There are 6 macrophage populations – where are they located and which one is fetal Hofbauer cells? The relative paucity of NK cells and abundance of macrophages suggests that these are not from decidua. But why are there so few blood/decidual T cells? The best way to resolve these issues is to sequence maternal blood (and cord blood if obtainable) at the same time and remove these from the analyses.

To resolve these issues, the authors should compare their immune and fibroblast datasets in S2-S3 with previous single-cell data from the first trimester (Suryawanshi et al. 2018, Vento-Tormo et al. 2018). The correspondence between the decidual macrophages described in this manuscript is unclear compared with the ones from previous studies. Surprisingly, the 3 dNK cells previously identified are not found in this new dataset.

Figure 1S3

The presence of lymphatics in decidua is disputed (Volchek 2010) and Hofbauer cells express LYVE1 (Bockle 2008) so how certain is the presence of lymphatics.

The authors integrate the data from the villous chorion and smooth chorion computationally (eg Figure 1b). However, they should demonstrate that the integration pipeline is working correctly by defining the same cell states identified in the joint manifold when analysing the data separately. The same applies to all the analyses when specific cell subsets are zoomed in. For example, Figure 2c.

In addition, for each of the UMAPs defined in the manuscript, it is important to colour them by individual to confirm the integration.

In Figure 2 the authors introduce a CTB cycling population. Is this population corresponding to CTB-1 only? Or, is it a mix of CTB-1-4? If the latter, defining independent clusters by, for example, removing cell cycling genes from the highly variable genes considered to define the clusters should be performed.

The EVT populations described are problematic and the logic of this section is hard to follow. Most trophoblast cells in the SC express the definitive EVT marker, HLA-G (Hutter 1996). So how do these EVT subsets relate to CTR2-4 and to published datasets? It is also surprising that the same EVT populations are defined in the analysis of villous and smooth chorion. An artifact due to the computational algorithm could be forcing the integration and it needs to be shown that this is not the case. This is common, as by default, the method will assume equivalent populations in both datasets. One way to do so may be by using alternative methods for data integration (e.g., harmony, scVI), or by analysing datasets independently and comparing them afterwards (e.g. correlation analysis, machine-learning tools). It is also not made clear that the interstitial trophoblast deep in the decidua have not been sampled and only those EVT in the cell columns.

Line 368 "within the arterial wall, this population (interstitial EVT) further differentiates to endovascular EVTs and replaces the maternal arterial endothelium (Harris 2009, Red-Horse 2004)". This is controversial with the majority view instead being that interstitial trophoblast mediates medial destruction and then there is the replacement of the endothelium by endovascular trophoblast moving down from the shell (Pijnenborg; Bulmer).

Line 417. The "suggesting a cell non-autonomous, rather than cell-autonomous, regulation of invasion" is speculation that should be included in discussion rather than the results. How do the authors explain the fact that trophoblast organoids invade in the absence of decidua as in tubal or abdominal pregnancy? To increase the impact the proteomes of the two conditioned media from VC and SC should be studied to identify what is inhibiting invasion into decidua.

---

## [Author Response]

Essential revisions:Overall, this study is novel and addresses an important and uninvestigated question utilizing scRNA-seq and elegant computational approaches. The reviewers consider that the field will benefit from the publication of this research. Yet, there are caveats that the authors need to address before publication.General comments:(1) A point raised by both Reviewers #1 and #3 is about sample collection. We suggest the authors explain in detail in the method sections how the cellular isolations of VC and SC trophoblasts were performed and not just cite other papers. In addition, it would be good to know how the samples were obtained. Were they normal and voluntarily terminated? Reviewer #3 mentions that only very rarely do pregnancies fail in the second trimester which is a safe period for the mother and baby, so what is the reason for their delivery? If they were voluntary terminations of pregnancy they will have been exposed to mifepristone/misoprostol that will affect gene expression. Whichever way they were obtained they are not normal. Were the samples of the SC taken from close to the disc or near the cervix?

We thank the reviewers for this suggestion. We have added more detailed protocols for dissection and cellular isolation from both the villous and smooth chorion to the methods section. All samples were healthy and voluntarily terminated in the second trimester between weeks 17 and 24. This information has been added to the methods section.

We also thank the reviewers for the consideration of the influence of the elective termination procedure on the cells we collected. Mifepristone and Misoprostol were not used in the termination of these pregnancies.

As to the location of the samples, the SC samples include cells isolated from varying distances from the disk and cervix. Briefly, the fetal membranes (often not fully intact) were washed and the amnion and decidual parietalis removed. The tissue was then minced into small pieces. These pieces contain tissue from many points along the SC. Therefore, the samples represent a sampling of cells from multiple locations in the SC, not biopsies of specific locations.

(2) Reviewers #1 and #2 raised a similar suggestion. The authors mention that there are differences in the EVTs isolated from the placental villi and those obtained from the chorioamniotic membranes. Other studies from the first and third trimesters have documented the presence of EVTs using scRNA-seq. I think that the study would benefit from comparing the authors' data (second trimester) with these publically available datasets. I think the authors could provide a unique roadmap of EVT transcriptomic activity throughout gestation.

We thank the reviewers for the suggestion of incorporating public data to strengthen our analysis of the trophoblast populations, including EVTs. We compared our trophoblast data to data from the VC of first trimester placentas from Vento-Tormo et al., 2018 and to data from the VC and SC of term placentas from Pique-Regi et al., 2019. First, we performed pairwise comparisons of the populations recovered in each of these studies, to identify which cell types were common among studies (Figure 2-S1d and f). This demonstrated the transcriptome of CTB 4 (and to a lesser extent CTB 3) not to be correlated with populations found in either Vento-Tormo et al., 2018 or Pique-Regi et al., 2019. Additionally, this analysis identified a strong correlation between the EVT populations identified in each dataset. EVT identified in Vento-Tormo et al., 2018 (first trimester) showed a stronger correlation with the less mature EVT populations in our dataset, while EVT identified in Pique-Regi et al., 2019 (term) demonstrated strong correlation with the most mature EVT clusters (2-4) in our data. To investigate this with higher resolution we integrated the data from each dataset with our own. Integration of the data from Vento-Tormo et al., 2018 show first trimester EVTs (including cells FAC sorted from HLA-G protein expression) align with our EVT 1 population. This suggests these first trimester EVTs resemble the least mature EVTs in our data and lack several features of invasive cells. Integration of the data from Pique-Regi et al., 2019 show strong similarity to those identified in our dataset. The cells isolated from the SC at term appear restricted to the less invasive EVT clusters in our dataset. Together these analyses suggest that EVTs in the SC may decrease an invasive transcriptional program between the second trimester and term. However, as there are notable differences in the populations recovered by Pique-Regi et al., 2019 and our study, including SC-CTBs (not found in Pique-Regi et al., 2019) and npi-CTBs (not found in our data), we cannot rule out that differences in EVT transcriptional signature may not be solely due to gestational age but due to differences in dissection and. Therefore, we are cautious about expanding claims about the relationship between EVTs and developmental time beyond these analyses.

(3) The findings on SC-CTB-secreted factors inhibiting EVT invasion are very intriguing. Could the authors analyze the conditioned media (via mass spec or ELISA) from SC and VC to try to identify potential candidates (e.g. SERPINE1 as mentioned in the discussion)?

We agree with the reviewers that the potential for specific secreted factors from SCCTBs suppressing EVT migration is exciting and that their discovery would be exceptionally impactful. Indeed, Dr. Blelloch is currently preparing a grant proposing to seek and study such factors including by mass spec and ELISA panels. This grant proposal is a five-year plan and we feel beyond the scope of the current manuscript.

(4) A major concern raised by Reviewer #2 is about the nomenclature used in the paper: villous chorion and smooth chorion. According to the schematic representation shown in Figure 1A, the authors sample the placental villous attached to the basal plate. Is this correct? It would have been more appropriate to sample the placental villous neighboring the chorionic plate or that from the inner placental mass, which is not associated with the basal or chorionic plate.

We apologize to the reviews for the lack of clarity around the regions that were sampled in this study. We sampled cells from both floating villi and anchoring villi. We sampled these regions to ensure capture of CTBs, STBs, and EVTs. A schematic of the anchoring villi was provided because it depicts all of these cell types, whereas a schematic of the floating villi would not depict EVTs. We also deem it necessary to highlight the villous cell types including invasive EVTs in the VC to contrast with the SC trophoblast epithelium which lacks both villi and invasive EVTs. The text has been updated to specify that both floating and anchoring villi were collected.

We have included a more detailed explanation of the method of sampling. Cuts were made at the base of each cotyledon near the chorionic plate and the entire villous tree up to and including the basal plate was taken for study. The decidua was removed from the basal plate side and the remaining villous tree was dissected and dissociated. This retained cells in the basal plate (including cells moving through the villous cell column and beginning invasion into the decidua) and the floating villi near the chorionic plate and the inner placental mass.

In addition, the names utilized to define these placental compartments are unconventional. The readers would benefit from the authors utilizing conventional nomenclatures such as placental villi or chorioamniotic membranes. Last, the trophoblast in the chorioamniotic membranes is termed chorion laeve. The authors should consider using such terms throughout the manuscript and figures.

We appreciate the reviewers concern over the nomenclature in the manuscript. The terms villous chorion and smooth chorion are accepted and widely used in publications (Benirschke et al., 2006; Genbacev et al., 2015; Genbacev et al., 2016; Yuan et al., 2006; Yuan et al., 2008; Yuan et al., 2009; King, 1981; Garrido-Gomez et al., 2017). While we appreciate that placental biologists and clinicians still use the traditional Latin names of chorionic villi and chorion laeve. Since we are submitting this study to *eLife* for a general audience we would like to maintain the use of common English terms for the clarity of the general readership. We have rewritten the introduction to clarify the nomenclature we use and put it in the context of the more specialized terms such as chorioamniotic membranes.

(5) Recent studies have suggested that stromal cells from the membranes participate in the host response against viral infection (COVID-19). Did the authors analyze the different types of stromal cell types between the placental villi and the membranes, and whether these display shared or unique functionality? Please discuss.

We thank the reviewers for this suggestion. We feel it would be off topic and inappropriate to comment on the application of these data to the ongoing COVID-19 pandemic for multiple reasons. Primarily, the trophoblast enrichment protocol specifically selects against stromal cells. We recover fewer than 1000 cells from VC samples, including two samples with just 1 and 13 cells. This analysis is deeply underpowered to identify differentially expressed markers with specific functions in viral infection. Future cellular isolations which do not deplete stromal cells will be necessary to answer this question. Second, there are several datasets which have better representation of the stroma which would be better suited to answer this question (Pique-Regi et al., 2019; Pique-Regi et al., 2020; Garcia-Flores et al., 2022). Lastly, our study was designed to focus on trophoblast cell fate and function and a deviation to explore anti-viral properties of the stromal cells is beyond the scope of this work.

(6) Overall, the study would benefit from incorporating publically available data into the manuscript to strengthen the conclusions.

We thank the reviewers for these suggestions. We have incorporated two publicly available datasets, Vento-Tormo et al., 2018 (an atlas of first trimester villous chorion cells including emphasis on the stroma and immune populations) and the Pique-Regi et al., 2019 (a dataset that profiles both the villous and smooth chorions at term). First, we compared the cell types we identified in each of the trophoblast, stroma, and immune subsets to the cell types identified in Vento-Tormo et al., 2018 (Figure 1-S3a and c; Figure 2-S1d) and in Pique-Regi et al., 2019 (Figure 1-S4a and c; Figure 2-S1f). We found strong correlations in average expression between our cluster and those previously identified, which allowed for the validation of our cell type annotations. This comparison did not involve any computational integration and demonstrated the recovery of similar cell types in independent clustering analyses. Second, we computationally integrated our data with Vento-Tormo et al., 2018 (Figure 1-S3b and d; Figure 2 S1e) and Pique-Regi et al., 2019 (Figure 1-S4b and d; Figure 2-S1g), which demonstrated similarity between the cell types identified in each dataset.

(7) Differences between CTBs and EVTs can also rely on cell-cell communications. Therefore, the authors are encouraged to perform Cell-Cell Chat analysis using their data.

We thank the reviewers for the suggestion to model receptor-ligand interactions between cell types in each region. We analyzed the predicted signaling interactions within cell population isolated from each region using CellPhoneDB and focused on three comparisons: CTB-CTB interactions (VC and SC), EVT-stromal interactions (VC and SC) and, CTB-EVT interactions (only in SC).

Since the four distinct CTB populations in the SC reside in close proximity to each other, we hypothesized that signaling between these populations may instruct the differentiation from CTB 1 to CTB 4 (SC-CTB). The number of predicted receptor-ligand interactions unique to the SC between CTB populations far outnumber those unique to the VC, which is largely composed of only CTB 1 cells. We plotted the predicted interactions unique to each region and the weight of the interactions estimated by mean expression of both receptor and ligand in the heatmaps in Figure 3-S2. Several pathways were unique to the SC including BMP and Notch signaling, as well as, several members of the EPH-EPHRIN family (Figure 3-S2b). These interactions may contribute to the function of the SC through influencing cell fate and/or cell sorting.

Next, we analyzed the predicted interactions between EVT and stromal cells within each region to identify differential signaling events that might contribute to reduced invasion in the SC. We plotted the predicted interactions unique to each region and the weight of the interactions estimated by mean expression of both receptor and ligand in the heatmaps in Figure 5-S3. We identified a greater number of interactions unique to the VC than the SC, and these interactions were enriched for Collagen-Integrin signaling from the EVTs to stromal cells (Figure 5-S3a). Interactions unique to EVT-stroma in the SC included PDGFA and FN1 secreted by EVTs to receptors various stromal cells

(Figure 5-S3b). Future work will test the functional consequences of these interactions.

Finally, since we observed frequent CTB-EVT interactions in the SC, we investigated signaling between these populations. We did not perform this analysis for the VC cells as these populations do not contact each other in the VC and any predicted interactions would be false-positives. We plotted the predicted interactions between CTBs and EVTs in the SC and the weight of the interactions estimated by mean expression of both receptor and ligand in the heatmap in Figure 6-S4. To attempt to identify secreted factors which might be responsible for the reduced invasion in VC cells upon treatment with SC conditioned media (Figure 6d), we analyzed the interactions between CTB 4 and EVTs which contained a secreted factor (Figure 6e). This analysis revealed numerous factors for future study, including FN1, TNFa, PGF, THBS1, TGFB1, FGF1, and PDGFB.

(8) The manuscript is clearly presented and well written with a comprehensive reference to primary papers. However, many cited references are not included in the list including Maltepe 2015; Boll-Resli 1981; Pique-Regi 2020; Schlegelmilch 2011; Turco 2018; Kim 2006; Garcia-Flores et al. 2022.

We apologize for the oversight in not providing appropriate citations. All citations are now added.

(9) Reviewer #3 mentioned that identification of phenotypic markers specific to the smooth chorion is of interest and explains how this layer mediates important barrier functions for the fetus. An interesting finding is the specific expression of KRT6A in the smooth chorion. What is not clear from the report is how this layer is generated from the villi that originally surround the conceptus, which then regress probably due to the high oxygen concentrations at the periphery (Burton). This process is defective when the trophoblast transformation of vessels is abnormal in conditions, such as pre-eclampsia and preterm labour. The findings would be considerably more interesting if they could access samples earlier in pregnancy as the underlying pathogenesis of preterm labour is still unclear.Thus, the important unanswered question is: what in these two microenvironments is driving the differential trajectory of CTB 1 towards villous SCT or to smooth KRT6A+ cells in the smooth chorion? Some TFs are different but what causes their upregulation in SC and this needs addressing?

We thank the reviewers for their interest and this insight. Like the identification of specific factors that explain the suppression of EVT migration on SC side would be very exciting, so would be the uncovering of the basis of differential trajectories for CTB1 differentiation on the two sides. As suggested here, it very well may be differential oxygen tension, but there are countless other possibilities. It may even be an underlying cell autonomous factor such as epigenetic differences that we cannot see at the RNA level. Looking earlier, i.e. the first trimester, might be one way to get at this question, although prior to clear morphological differences of the two sides, it will be difficult to know what will become future villous vs. smooth chorion from the sort of surgical specimens we receive from the clinic. In all, these are exciting, but complex questions with likely complex underlying mechanisms. We certainly plan to pursue these questions, but again feel it is beyond the scope of the paper. We believe the discovery of the different trajectories and of a previously undescribed CTB population on the SC side is already an important advance in the field. We hope the reviewers agree.

Are there any cells with the SC transcriptome in the new models of human TSC (Okae 2018; Turco 2018; Haider 2018)? These provide an in vitro model to generate SC and would considerably strengthen the paper.

We thank the reviewers for the suggestion to analyze data from existing culture systems to identify if SC-CTBs are present. The three resources for placental organoid culture (Okae 2018; Turco 2018; Haider 2018) do not perform scRNA-seq which is a limitation for detecting rare subpopulations of cells. These studies performed bulk RNA analyses and we investigated whether KRT6A transcript was expressed in any of these datasets.

Okae et al., 2018, performed bullk RNA-seq to compare the transcriptional profiles of cells isolated from chorionic villi, human trophoblast stem cells, and differentiated cells from human trophoblast stem cells. They identified very low average KRT6A transcript in cells isolated from tissue [CTB = 0.018, EVT = 0.194, STB = 0.016 log2(FPKM+1)], in accordance with the low level of expression in the VC in our study. In most cultured cells KRT6A transcript was not detected. A very low level of expression was detected in TS cell derived from blastocysts [0.016 log2(FPKM+1)] and in the 3D differentiation of STB from blastocysts [0.106 log2(FPKM+1)]. Therefore, we believe that SC-CTB are not generated by these culture methods or if they are, it must be in very small numbers.

Turco et al., 2018 performed RNA microarray comparing GW6-GW12 chorionic villi samples to cultured placental organoids. Average normalized expression of KRT6A does not change between tissue samples (2.16) and organoid samples (2.41). As we have established that KRT6A expressing cells are not found in the villous chorion, no increase in expression in organoids suggested SC-CTB were not present.

Haider et al., 2018 performed RNA-seq on CTB-organoids and compared transcriptional profiles to isolated CTB, STB, and fibroblasts. The average expression of KRT6A across all samples as calculated by DESeq2 baseMean was 69.7 compared to highly expressed CTB marker genes like KRT7 (4406) and PAGE4 (14392), ranking KRT6A expression 9740th out of 17008 transcripts identified. Additionally, KRT6A was not found to be differentially expressed in any comparison across conditions. Therefore, we concluded that SC-CTB are not generated in this organoid culture system.

We agree with the reviewers that the ability to derive SC-CTB in culture would greatly enhance our ability to study these cells. However, it seems these cells are not currently generated in either directed differentiation or heterogenous organoid culture models. We do hope in the future to be able to identify in vitro differentiation conditions that do produce SC-CTBs.

(10) The methods of cell isolation rely on referring back to a previous publication and need expanding. "CTBs were enriched over stromal and immune cells – line 126" needs explanation as this will be a potential source of bias in the results. It appears that samples of the maternal side of the main villous placenta have been obtained – is this correct? Was decidua deliberately included? For the smooth chorion was the whole chorion – amnion, stroma, trophoblast and decidua included? These details are important as selecting specific areas and removing others will obviously alter the comparisons made.

We thank the reviewers for the suggestion to expand details on the methods of dissection and cellular isolation. Please also see the response to point 1, as this raises the same question. We have greatly expanded the methods section for dissection and isolation for both VC and SC samples.

The goal of this study was to focus on the trophoblast populations. In the VC samples, floating villi and anchoring villi were carefully dissected away from the decidua. These dissections, not surprisingly, do retain some decidual cells, presumably those most closely associated with the cell columns. The cell preparation further enriches for trophoblast cells over the stroma cells (both decidua and fetal derived) as well as immune cells. The result is that stromal cells are not even found in all VC samples and the total number of stromal cells is less than 1000. Immune cells are found in greater numbers, but also likely reduced. In SC, both the decidua and amnion were dissected away. The remaining cells were once again prepared to enrich for trophoblasts over stromal and immune cells. The purposeful depletion of stromal and immune cells is now explicitly stated in the text. We only briefly mention the stromal and immune cell types we recover and make limited claims concerning their functions.

(11) There are 6 macrophage populations – where are they located and which one is fetal Hofbauer cells? The relative paucity of NK cells and abundance of macrophages suggest that these are not from decidua. But why are there so few blood/decidual T cells? The best way to resolve these issues is to sequence maternal blood (and cord blood if obtainable) at the same time and remove these from the analyses.To resolve these issues, the authors should compare their immune and fibroblast datasets in S2-S3 with previous single-cell data from the first trimester (Suryawanshi et al. 2018, Vento-Tormo et al. 2018). The correspondence between the decidual macrophages described in this manuscript is unclear compared with the ones from previous studies. Surprisingly, the 3 dNK cells previously identified are not found in this new dataset.

Again, this study focuses on the trophoblast cells at the expense of the stroma and immune cells. The reviewer is correct that the best way to identify the maternal cells is to sequence maternal blood (as is performed in Vento-Tormo et al. 2018). However, we did not do this because we sought to minimize the contribution of stromal and immune cells and to focus on the trophoblast.

Previous publications, like Vento-Tormo et al. 2018, specifically enriched for immune cells by both collecting peripheral blood and decidual samples, then sorting for CD45+ cell by FACS and performing scRNA-seq. Additionally, characterization of the three dNK cells mentioned by the reviewer includes data from the Smart-Seq2 method of scRNAseq. Because this method captures full length RNA and sequences to a much greater depth, Smart-Seq2 is a much more sensitive assay than droplet based scRNA-seq methods like 10x genomics. For these reasons, we do not find it surprising that our droplet based scRNA-seq data isolated from samples by a method specifically designed to exclude the decidual regions does not recover these cell states with the same resolution.

By design our experiment targets trophoblast cells at the expense of immune cells and stromal cells. We understand and admit the strengths and caveats of this approach, namely that there are populations missing from the immune and stromal cells. We have added language to further emphasize these caveats and our logic in the text.

To better understand the immune populations in our dataset and to compare to existing cell type definitions we have compared the immune subsets from this study to published data in Vento-Tormo et al., 2018 and Pique-Regi et al., 2019. We did this both through correlation of expression between independently derived clusters and through the computation integration of these datasets. Our NK cells show strong correlation and integration with the three dNK subsets identified in Vento-Tormo et al., 2018 (Figure 1S5e and f).

Based on the comparison of our macrophage populations to those identified in Vento-

Tormo et al., 2018 we have amended the cluster annotations slightly. The new Macrophage 1 (M) cluster matches the dM1 cluster from Vento-Tormo et al., 2018 and contains the old Macrophage 1 (M), Macrophage 2.1 (M), Macrophage 2.2 (M) clusters. The new Macrophage 2 (M) cluster matches the dM2 cluster from Vento-Tormo et al.,

2018 and contains the old Macrophage 2.3 (M) and Macrophage 2.4 (M). Macrophage 3 (M) matches the dM3 cluster from Vento-Tormo et al., 2018 and remains unchanged. These changes are updated in Figure 1-S5.

(12) The authors integrate the data from the villous chorion and smooth chorion computationally (eg Figure 1b). However, they should demonstrate that the integration pipeline is working correctly by defining the same cell states identified in the joint manifold when analysing the data separately. The same applies to all the analyses when specific cell subsets are zoomed in. For example, Figure 2c.

We thank the reviewers for raising this point. To address this, we analyzed the cells from the VC and SC independently as suggested. Focusing on the trophoblast cells, clustering identified similar clusters as the integrated dataset. Independent analysis of the VC identified all clusters except for CTB 4 (the smooth chorion specific population), suggesting that the small number found in the integrated analysis were coerced by cellular integration. Analogously, independent analysis of the SC identified all clusters except for STB Precursor, STB, and EVT Precursor cells. These results are consistent the small number of STB (14 cells) and EVT precursor (8 cells) from the SC in the integrated dataset (Supplementary Table 1). We have included these analyses in an additional supplementary figure (Figure 2-S3a-f). We performed the same analyses for both the stroma and immune subsets and included these analyses in new supplementary figures (Figure 1-S6a-d and Figure 1-S8a-d).

In addition, for each of the UMAPs defined in the manuscript, it is important to colour them by individual to confirm the integration.

We thank the reviewers for this suggestion. The contributions of each individual to each UMAP colored by coarse cluster are provided in Figure 1-S1c and the contribution of each individual to the trophoblast subset colored by cluster annotation is shown in Figure 2-S1b. We have added the contribution of each individual to the immune and stroma subset UMAPs in Figure 1-S5c and Figure 1-S7c, respectively.

(13) The EVT populations described are problematic and the logic of this section is hard to follow. Most trophoblast cells in the SC express the definitive EVT marker, HLA-G (Hutter 1996).

We regret the confusion in this section. While we appreciate that prior work has suggested that most trophoblast cells in the smooth chorion express HLA-G (Hutter, 1996), we do not find this to be true. In our scRNA-seq data we identify that only 46.8% of trophoblasts in the VC and 32.1% of trophoblasts in the SC express HLA-G (Figure 5c). The mix of trophoblast population in SC was further supported by staining for HLAG protein in the SC (Figure 5b, Figure 6b, Figure 6-S1). Our findings are also supported in the literature by HLA-G staining (Garrido-Gomez et al., 2017).

We agree with the reviewer that EVTs are marked by the definitive marker HLA-G. Expression of HLA-G transcript was used to define the EVT clusters we identified (Figure 2 and Figure 5) and protein expression of HLA-G is used to define EVT cell identity in all tissue sections (Figure 5 and Figure 6).

So how do these EVT subsets relate to CTR2-4 and to published datasets?

CTB 1-4 and CTB S-phase and CTB G2/M-phase cells do not express HLA-G transcript, as shown in Figure 5a and Figure 2-S2b. Additionally, we demonstrated that the CTB 1 marker np-CTNNB1 (Figure 6a and Figure 6-S1) and the CTB 4 marker KRT6A (Figure 6b and Figure 6-S1) have mutually exclusive domains of protein expression from EVT marker HLA-G. Therefore, our data support that EVT identified in either the VC or SC are distinct from all CTB populations (including all proliferating cells).

We compared the trophoblasts populations we identified to those published in VentoTormo et al., 2018. The EVT cells we identify bear strong resemblance to both computationally annotated EVT cells and FACS sorted HLA-G+ cells identified in VentoTormo et al., 2018 (Figure 2S^-1^d and e). In accordance with our data, all EVT and HLAG+ cells cluster separately from all CTB (including the smooth chorion specific CTB 4 cells) [Figure 2-S1e]. This provides evidence from an external dataset that the EVT we identify match previously published EVT and are distinct from all CTB isolated from either the VC or the SC.

It is also surprising that the same EVT populations are defined in the analysis of villous and smooth chorion. An artifact due to the computational algorithm could be forcing the integration and it needs to be shown that this is not the case. This is common, as, by default, the method will assume equivalent populations in both datasets. One way to do so may be by using alternative methods for data integration (e.g., harmony, scVI), or by analysing datasets independently and comparing them afterwards (e.g. correlation analysis, machine-learning tools).

We thank the reviewers for this suggestion. To address the possibility that similarities between EVT population are an artifact of integration we performed independent analysis of the cells isolated from each region (see the response to point 12). We identify both strong correlations and overlapping computational integrations between independently identified EVT clusters in the VC and SC samples (Figure 2-S3a-f), between the EVT populations identified in this study with EVT identified by Vento-Tormo et al., 2018 (defined by both by transcript and HLA-G protein expression)[Figure 2-S1d and e], and between the EVT populations identified in this study with EVT identified by Pique-Regi et al., 2019 (Figure 2-S1f and g). Please see the response to point 2 above for more detail. Given the agreement in the findings of multiple orthogonal methods and multiple independent datasets, we do not believe the transcriptional similarity between EVT from the VC and SC is an artifact of computational integration.

It is also not made clear that the interstitial trophoblast deep in the decidua have not been sampled and only those EVT in the cell columns.

We apologize for the lack of clarity around this point. We have amended the text and methods to make this clearer.

Please see lines 115-119: “VC samples included cells isolated from floating and anchoring villi and areas surrounding the cell column, while most of the decidua (including spiral arteries) were dissected away. SC samples included the chorion and underlying stroma (mesenchymal and endothelial cells), but not the amnion and little of the neighboring decidua, which were also removed during dissection”

Detailed comments:(1) Line 368 "within the arterial wall, this population (interstitial EVT) further differentiates to endovascular EVTs and replaces the maternal arterial endothelium (Harris 2009, Red-Horse 2004)". This is controversial with the majority view instead being that interstitial trophoblast mediates medial destruction and then there is the replacement of the endothelium by endovascular trophoblast moving down from the shell (Pijnenborg; Bulmer).

We thank the reviewers for highlighting the discussion around the source of endovascular EVTs. We have amended the text to reflect that there is not yet a definitive answer to this question and address both hypotheses. Please see lines 378381 – “The relationship between this subpopulation and endovascular EVTs that replace the maternal arterial endothelium is unclear, with evidence supporting endovascular EVTs arising from either from interstitial EVTs and/or an independent origin (Harris et al. 2009; Red-Horse et al. 2004; Pijnenborg et al., 2011).”

(2) Introduction: The second paragraph of the introduction could be rewritten and/or incorporated into the third paragraph. In its present form, it reads like a textbook. In addition, the introduction is somewhat too long and would benefit from some trimming.

We thank the reviewers for this helpful feedback on length and clarity. We have rewritten the introduction removing paragraph 2 as well as trimming other parts. However, given request made in point 4 of general comments, we also had to add text to clarify nomenclature we use throughout, so in the end the introduction is not much shorter than the original size.

(3) Introduction: The last paragraph of the introduction should be revised. The authors mention that previous studies using scRNA-seq were performed in the fetal membranes from term pregnancies (Page 3, Lines 104-105). Yet, no references were provided. In the following sentence (Page 3, Lines 105-106), the authors mention "in this study, CTBs were not identified in the smooth chorion…" and the authors cite four papers. Which of these papers are the authors referring to? In the 2019 study that is cited, the authors reported the presence of CTBs and EVTs in the fetal membranes. In addition, new studies have also reported the presence of CTBs and EVTs in the fetal membranes (PMIDs: 32662421, 35042863). The authors may consider including this information.

We apologize to the reviewers for lack of clarity in this section. Pique-Regi et al., 2019 was cited as a reference for the claim that CTBs were not identified in the smooth chorion. No claims were made about a lack of EVTs in either Pique-Regi et al., 2019 or in our manuscript. In Pique-Regi et al., 2019 the authors identify 132 CTBs (CTB and npi-CTB) out of 29,921 cells isolated from the chorioamniotic membranes, representing 0.44% of all cells recovered. We have also included reference to Garcia-Flores et al., 2022 as another dataset profiling the SC which contains a similar underrepresentation of CTBs. We have updated the text with accurate numbers and proper citations.

(4) Results, Page 3, Lines 122-124: The authors mention that they chose to analyze second-trimester samples to avoid inflammation and apoptosis associated with parturition. Couldn't the authors include samples from women who delivered at term without labor?

We thank the reviewers for mentioning these datasets. We have cited Garcia-Flores et al., 2022 in the manuscript as another scRNA-seq data set profiling the SC. Pique-Regi et al., 2020 uses the chorioamniotic Membrane data from Pique-Regi et al., 2019 and includes no new data from the chorioamniotic membranes and, while interesting and important, is not relevant to this study.

(5) Page 8, Lines 265-266: What is the evidence to suggest that the differentiation of SC-CTB does not require a cell cycle?

We thank the reviewers for this suggestion. Given the large changes observed in both the transcriptome and epigenome between the second trimester and term, we expect this to be a complete project unto itself. In the future, samples from placentas collected from delivery at term without labor will be considered to address the composition and roles of VC and SC trophoblast at term.

(6) Results, Page 5, Lines 157-159: The authors suggest the involvement of maternal immune cells in VC and SC; yet, these results are expected since both the placental villous and the chorioamniotic membranes (which were sampled in this study) represent the maternal-fetal interface: intervillous space and decidua parietalis, respectively.

We thank the reviewers for the opportunity to clarify this point. We do not claim (or have evidence) that the differentiation to CTB 4 does not require a cell cycle. We demonstrate KRT6+ CTB 4 cells express KI67, suggesting these cells have not exited the cell cycle. We cannot determine and do not comment on the intersection of cell cycle and differentiation beyond noting that CTB 4 cells can remain proliferative. Since both other differentiated cell types in the placenta (EVTs and STBs) exit the cell cycle in concert with differentiation (unlike CTB 4), we feel this is a distinction worth noting. We have updated the text to clarify this point.

Please see lines 268-270: “Mitotic KRT6+ cells were identified, and while the interaction between the cell cycle and differentiation of SC-CTBs remains unclear, these data show that differentiation to SC-CTB does not require cell cycle exit, unlike STBs and EVTs”

(7) Results: The authors are congratulated for validating the mRNA results with protein expression. Specifically referring to Figure 2. The authors are also congratulated for testing functional differences between the EVTs from the placental villi and those from the membranes. I think that this is a very useful piece of information, and the functionality of the different subsets of EVTs requires further investigation.

We thank the reviewers for these kind words. We look forward to investigating the functional heterogeneity of EVT subpopulations in the VC and SC in future studies.

(8) Results, Page 8, Line 245: The authors claim that the CTBs from the membranes and those from the placental villi share a common progenitor by performing an elegant RNA velocity analysis. Yet, wouldn't it be expected that all trophoblasts share a progenitor?

While it is not surprising that all trophoblasts derive from a common progenitor early in development (first trimester), it was not known whether a common progenitor for trophoblasts in two physically distinct regions remained well into the second trimester. The result we found surprising is that a common progenitor exists as late as GW23, this progenitor is transcriptionally similar to canonical Villous CTB (now demonstrated to exist in the SC), and that this progenitor appears fate restricted by its location.

(9) Figure 5: What is the evidence that EVTs from the two different compartments are behaviorally distinct? In other words, what do the authors mean by "behavioral"?

We thank the reviewers for pointing out the vagaries of this wording. We use the term behavioral to refer to the migratory/invasive actions of these cells in the tissue. In referencing distinct behavior, we refer to the documented differences in the depth of invasion/migration into maternal cells observed in the VC or the SC (Genbacev et al., 2015). To reduce confusion, we have changed “behavior” to “invasive activity”.

(10) Immune cells Figure 1S2How were the 4 different immune cell populations – maternal blood/decidual and fetal blood/placental cells distinguished? Both stromal and immune cells from both individuals will be present in these isolates. It is said by XIST that most of the immune cells are maternal – are they from blood or decidua? Were the fetuses all male? It would be better to look at the transcriptomes and deconvolute the genotype data by SNPs to confirm.

We thank the reviewers for pointing out the need for better characterization of the origin of stromal and immune cells. We agree with the reviewers that the best method to assess the origin of these populations is to collect blood and decidua and deconvolute by SNPs. Deconvolution by SNP from transcripts using programs like freemuxlet work well for separating reads from two distinct (unrelated) individuals but fail in the case of parental relationships due to shared genomic material. As this study was focused entirely on the trophoblast populations (fetal only), we did not sequence and genotype maternal decidua, maternal blood, and fetal only cells to perform this analysis.

Instead, by assessing XIST expression in trophoblast cells from each sample (all of fetal origin), we assigned the sex of each sample to be GW17.6 = Male, GW18.2 = Male, GW23 = Female, and GW24 = Male (Figure 1-S2a). Therefore, in the stromal and immune populations cells of fetal origin would only express XIST in GW23 samples and no others. All immune populations, with the exception of Erythrocytes, express XIST in all samples (Figure 1-S2c). Stromal populations with XIST expression largely restricted to GW23 samples (Mesenchyme 1 (F), Mesenchyme 3 (F), and Mesenchyme 4 (F)), are annotated as fetal. All others show XIST expression across all samples and are thus annotated as maternal (Figure 1-S2b).

(11) Figure 1S3The presence of lymphatics in decidua is disputed (Volchek 2010) and Hofbauer cells express LYVE1 (Bockle 2008) so how certain is the presence of lymphatics.

The lymphatic endothelium in the smooth chorion has been previously identified, validated, and localized in Pique-Regi et al., 2019. We identify strong transcriptomic correlations between the Endo Lymph cells we identify and those identified in PiqueRegi et al., 2019 and Vento-Tormo et al. 2018, demonstrating the lymphatics identified in three studies to be similar (Figure 1-S3a and b; Figure 1-S4a and b). While we did not perform any immunolocalization, Pique-Regi et al., 2019 did confirm the location of these cells in the SC using an antibody to LYVE1.

(12) In Figure 2 the authors introduce a CTB cycling population. Is this population corresponding to CTB-1 only? Or, is it a mix of CTB-1-4? If the latter, defining independent clusters by, for example, removing cell cycling genes from the highly variable genes considered to define the clusters should be performed.

We thank the reviewers for pointing out the lack of clarity around the identity of these populations. The S-phase and G2/M-phase CTB populations contain a mix of cells from CTB 1-4 (but not from STB, EVT, or either precursor populations). Although it can be beneficial to regress out cell cycle related genes to remove factors which obscure cell identity, we did not remove cell cycle genes from our analysis for two reasons. First, cell cycle is an important component of cell fate and identity. This is especially true in the placenta as both commitment to EVT and STB lineages is inextricably tied to cell fate (Lu et al., 2017; Genbacev et al., 1997). Therefore, we determined that excluding information relating to cell cycle state might decrease the ability to identify determinants of cell fate. The S-phase and G2/M-phase CTB containing cells expressing markers of CTB 1-4 clusters, motivated the analysis of proliferation by KI67 staining (Figure 4b and Figure 3-S1b) that demonstrates that CTB 4 cells remain proliferative. This sets this population apart from both EVT and STB. Second, the majority of CTB identified in our analysis do not contain markers of S or G2/M phase transcripts. Therefore, we are confident in the ability to define cell identity solely based on non-phasic transcripts in these 26,325 cells.

(13) Line 417. The "suggesting a cell non-autonomous, rather than cell-autonomous, regulation of invasion" is speculation that should be included in discussion rather than the results. How do the authors explain the fact that trophoblast organoids invade in the absence of decidua as in tubal or abdominal pregnancy?

We thank the reviewers for the opportunity to clarify this point. We have removed this from the Results section. We believe our data suggest that the default action of EVT is to invade/migrate and the environment (SC-CTB and decidual cells) may function to restrict this invasion. If invasion is the default, EVTs will invade in the absence of SCCTB and decidual cells, matching what is observed in organoid models. Therefore, we believe our model is consistent with what is seen in in vitro models and tubal or abdominal pregnancies.

To increase the impact the proteomes of the two conditioned media from VC and SC should be studied to identify what is inhibiting invasion into decidua.

See answer to question 3 in general comments.

References (not included in manuscript)

King, B. F. (1981). Developmental changes in the fine structure of the chorion laeve (Smooth chorion) of the rhesus monkey placenta. *The Anatomical Record*, *200*(2), 163–175. https://doi.org/10.1002/ar.1092000206

Hutter, H., Hammer, A., Blaschitz, A., Hartmann, M., Ebbesen, P., Dohr, G., Ziegler, A., and Uchanska-Ziegler, B. (1997). Erratum: Expression of HLA class I molecules in human first trimester and term placenta trophoblast (Cell and Tissue Research (1996) 286 (439-447)). *Cell and Tissue Research*, *287*(3), 625. https://doi.org/10.1007/s004410050786